# Forecasting Hurricane-forced Significant Wave Heights using the Long Short-Term Memory Network in the Caribbean Sea

Brandon J. Bethel[1], Wenjin Sun[1,2], Changming Dong[1,2,3*], Dongxia Wang[4]

[1] School of Marine Sciences, Nanjing University of Information Science and Technology, Nanjing 210044, China

[2] Southern Ocean Science and Engineering Guangdong Laboratory (Zhuhai), Zhuhai 519000, China

[3] Department of Atmospheric and Oceanic Sciences, University of California, Los Angeles, CA 90095, USA

[4] China State Shipbuilding Corporation (Chongqing) Haizhuang Windpower Equipment Co., Ltd., Chongqing 400021, China

*Correspondence to: Changming Dong (cmdong@nuist.edu.cn)

**Abstract.** A Long Short-Term Memory (LSTM) neural network is proposed to predict hurricane-forced significant wave heights (SWH) in the Caribbean Sea (CS) based on a dataset of 20 CS, Gulf of Mexico, and Western Atlantic hurricane events collected from 10 buoys from 2010 – 2020. SWH nowcasting and forecasting are initiated using LSTM on 0-, 3-, 6-, 9-, and 12-hr horizons. Through examining study cases Hurricanes Dorian (2019), Sandy (2012), and Igor (2010), results illustrate that the model is well suited to forecast hurricane-forced wave heights, but also much more rapidly, at a significantly cheaper computational cost as compared to numerical wave models, and much lower required expertise. Forecasts are highly accurate with regards to observations. For example, Hurricane Dorian nowcasts had correlation (R), root mean square error (RMSE), and mean absolute percentage error (MAPE) values of 0.99, 0.16 m, and 2.6%, respectively. Similarly, on the 3-, 6-, 9-, and 12-hr forecasts, results produced R (RMSE; MAPE) values of 0.95 (0.51 m; 7.99%), 0.92 (0.74 m; 10.83%), 0.85 (1 m; 13.13%), and 0.84 (1.24 m; 14.82%), respectively. In general, the model can provide accurate predictions within twelve hrs (R $\geq$0.8) and errors can be maintained at under 1 m within six hrs of forecast lead time. However, the model also consistently over-predicted the maximum observed SWHs. From a comparison of LSTM with a third-generation wave model, Simulating Waves Nearshore (SWAN), it was identified that when using Hurricane Dorian as a case example, nowcasts were far more accurate with regards to the observations. This demonstrates that LSTM can be used to supplement, but perhaps not replace, computationally expensive numerical wave models for forecasting extreme wave heights. As such, addressing the fundamental problem of phase shifting and other errors in LSTM or other data-driven forecasting should receive greater scrutiny from Small Island Developing States. To improve models results, additional research should be geared towards improving single-point LSTM neural network training datasets by considering hurricane track and identifying the hurricane quadrant in which buoy observations are made.

**Keywords:** hurricanes; significant wave height; wave height forecasting; Long Short-Term Memory network; Hurricane

30    Dorian; Small Island Developing States; Caribbean Sea

## 1.    Introduction

Ordinarily, momentum and mechanical energy are transferred to the ocean's surface from the overlying atmosphere, giving rise to the ubiquitous surface gravity waves. Under forcing by tropical cyclones (TC), these waves become extreme and pose significant risks to coastal communities. As such, the study of TC-induced extreme significant wave heights (SWH) is at the current forefront of research and is traditionally accomplished by using an array of numerical models (Shao et al., 2019; Chao et al., 2020; Hu et al., 2020). However, although hindcasting, nowcasting, and forecasting (Alina et al., 2019; Cecilio and Dillenburg, 2020) can be performed using these models, they are all disadvantaged in that they all require large investments in high-performance computing resources, technical and scientific expertise, and crucially, time. For the Small Island Developing States and coastal communities of the Caribbean Sea (CS) which have yet to significantly invest in numerical modeling capabilities, other computationally cost-effective measures are required for wave height predictions. Consequently, alternatives are high priority. Recent research into artificial intelligence (AI)-based methodologies have shown that these techniques are highly effective at forecasting wave properties with minor computational expense, even under TC-forced states (Qiao and Myers, 2020; 2021).

Demonstrating, Chen et al. (2021) constructed a random forest (RF) supervised learning classifier to generate a surrogate for the Simulating Waves Nearshore (SWAN) third-generation numerical model and reduced the required computational time by a factor of 100. Wu et al. (2020) considered a physics-based machine learning model in conjunction with an artificial neural network for predictions of SWH and peak wave period where wind forcing, and initial wave boundary conditions are considered as inputs. Campos et al. (2021) used RF to select wind and wave variables to enhance wave forecasts. They found that RF was able to select the best forecast only in very short ranges using inputs of SWH, wave direction and period. However, variable selection for longer forecasts (five days and above) was much less certain. Huang and Dong (2021) improved upon the short-term prediction of SWH by decomposing deterministic and stochastic components using a complete ensemble empirical mode decomposition (CEEMD) algorithm and recurrence quantification analysis. A similar study by Zhou et al. (2021a) demonstrated that combining EMD and the long short-term memory (LSTM) network could also reduce SWH forecasting errors in the CS.

These methods are also effective under TC conditions. Important for the present study, Chen et al. (2020) applied a machine learning method to perform probabilistic forecasting of typhoon-forced coastal wave heights and found that the model could, based on wave height data and an array of typhoon characteristics, generate the predicted confidence interval that enclosed observed wave heights. Meng et al. (2021) considered introducing a deep learning method for long-term predictions of TC-

forced nearshore wave heights. The bidirectional Gated Recurrent Unit network was identified as an effective model for real-time and 24-hrs ahead predictions. Wei and Cheng (2020) developed a two-step wind-wave prediction model to predict wind speed and wave height under typhoon conditions and compared results with a one-step approach. It was identified that deep recurrent neural networks could be used for forecasting in either case, but the two-step approach was more effective. Zhou et al. (2021b) used the convolutional-LSTM (ConvLSTM) network to predict TC-induced SWHs in the South China Sea and found that up to a 12-hr forecast horizon, the correlation between forecasted values and observations could reach 0.94.

Recently, Bethel et al. (2021a) used LSTM to eliminate gaps in either surface wind speed or SWH by using one variable as a predictand to forecast its counterpart. While mean states were the focus of that study, one hurricane was used to demonstrate the methodology's effectiveness under extreme states. This study continues along that path to generate an LSTM-based forecast model exclusively for hurricane-forced SWHs in the CS using a set of input variables. This is deemed important for assessing and mitigating the risk of catastrophic losses in life and economic productivity due to hurricanes as seen most recently with the September 1st, 2019, landfalling of Hurricane Dorian in The Bahamas.

The remainder of this paper is structured as follows. Section 2 describes the data and methodology employed. Section 3 presents the main findings of this study. Sections 4 and 5 provide a discussion and the conclusion, respectively.

## 2. Data and Methodology

### 2.1 Observational Data

This study employs 10 buoys located throughout the CS, Gulf of Mexico, and Western Atlantic Ocean (Figure 1; Table 1) that are owned and operated by the National Data Buoy Center (NDBC; https://www.ndbc.noaa.gov/). Acquired variables include observations of surface wind speed and SWH. Gaps in buoy observations were processed using the insertion of WaveWatch III reanalysis data acquired from the Pacific Islands Ocean Observing System (https://coastwatch.pfeg.noaa.gov/). A total of twenty hurricanes identified from 2010 – 2020 were used and split into LSTM training and test datasets (Table 2). Hurricane statistics were acquired from the hurricane database maintained by the National Hurricane Center (https://www.nhc.noaa.gov/).

**Table 1. List of National Data Buoy Center buoys and their statistics.**

| Buoy No. | Buoy ID | Latitude (°N) | Longitude (°W) | Anemometer Height (m) | Water Depth (m) |
|---|---|---|---|---|---|
| 1 | 42002 | 26.055 | 93.64 | 3.8 | 3088 |
| 2 | 41010 | 28.878 | 78.485 | 4.1 | 890 |

| | | | | | |
|---|---|---|---|---|---|
| 3 | 41043 | 21.030 | 64.790 | 4.1 | 5362 |
| 4 | 41046 | 23.822 | 68.384 | 3.8 | 5549 |
| 5 | 41047 | 27.514 | 71.494 | 3.7 | 5321 |
| 6 | 41048 | 31.831 | 69.573 | 4.1 | 5394 |
| 7 | 41049 | 27.490 | 62.938 | 4.1 | 5459 |
| 8 | 42056 | 19.820 | 84.945 | 4.1 | 4554 |
| 9 | 42057 | 16.908 | 81.422 | 3.8 | 377 |
| 10 | 42058 | 14.776 | 74.548 | 3.8 | 4100 |

In some cases (e.g., Earl (2010), Igor (2010), Dorian (2019), Delta (2020)), the same hurricane was observed multiple
times along its track. To increase the total length of the LSTM training/test sets, these data segments were arranged into a
single time series. Additionally, cases such as Hurricane Humberto (2019) were explicitly excluded as swell contamination of
the wave field could potentially lead to poor forecasts, despite its classification as a major hurricane, large effects on the marine
environment (Avila-Alonso et al., 2021), and damage to the British overseas territory of Bermuda. Indeed, when a recently
developed empirical wind-wave model for the CS was applied to Hurricane Humberto (2019) by Bethel et al. (2021b),
observations of wind speed was a very poor predictor of the wave height and thus, given that surface wind speed and SWH are
being used jointly here, worsening of LSTM predictions using Hurricane Humberto (2019) in the training dataset is natural.
Unfortunately, it may not be possible to know a priori the existence of swell that may interfere with linear wind-wave
relationships and as thus, this is a disadvantage of the current model.

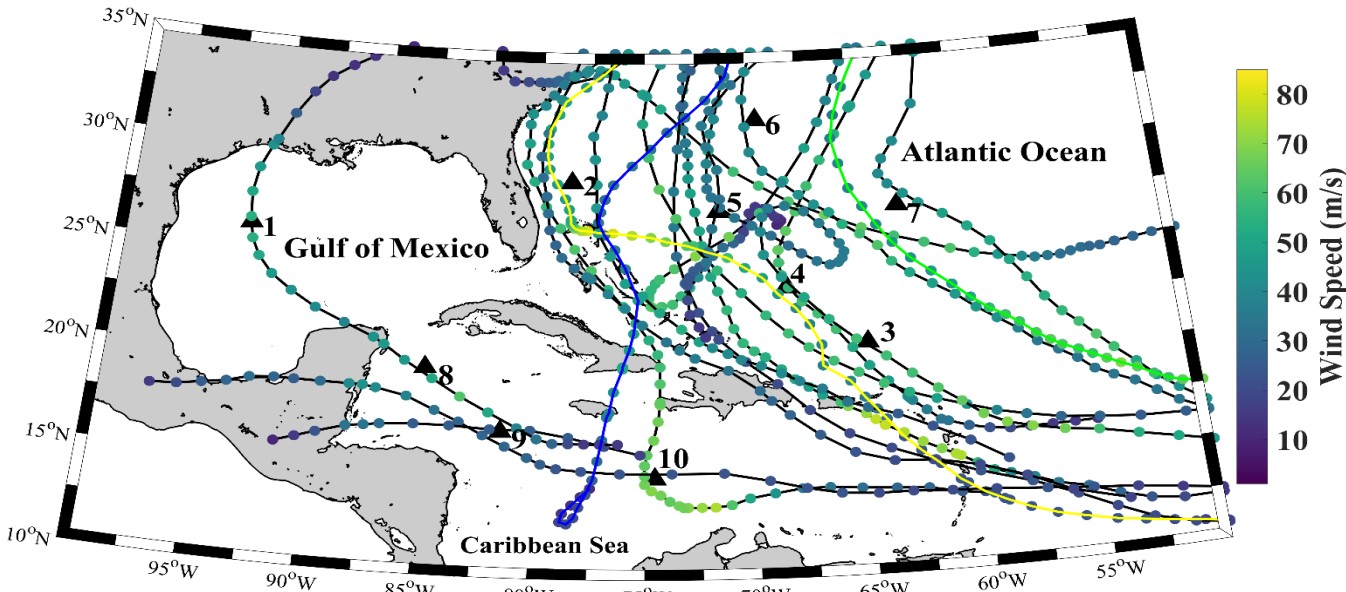


**Figure 1. Geographic map of the Caribbean Sea, Gulf of Mexico, and Western Atlantic Ocean with the best-tracks of each studied**
**hurricane and National Data Buoy Center (NDBC) buoy locations (black triangles). Best-tracks from model training hurricanes are**

given in black, while the test best-tracks are given in yellow, blue, and green for Hurricanes Dorian, Sandy, and Igor, respectively. Numbered from 1 – 10, the NDBC buoys employed are buoys 42002, 41010, 41043, 41046, 41047, 41048, 41049, 42056, 42057, and 42058, respectively.

Table 2. Formation/dissipation dates, minimum air pressures and maximum wind speeds of the twenty hurricanes used in this study.

| Dataset | Hurricane (YYYY) | Formation Date (MM/DD) | Dissipation Date (MM/DD) | Minimum Air Pressure (hPa) | Maximum Wind Speed (m/s) |
|---|---|---|---|---|---|
| Training Set | Earl (2010) | 8/25 | 9/5 | 927 | 63.8 |
| | Irene (2011) | 8/21 | 8/30 | 942 | 54.16 |
| | Katia (2011) | 8/29 | 9/12 | 942 | 61.1 |
| | Ernesto (2012) | 8/1 | 8/10 | 973 | 43 |
| | Cristobal (2014) | 8/23 | 9/2 | 965 | 38.8 |
| | Gonzalo (2014) | 10/12 | 10/20 | 940 | 63.8 |
| | Bertha (2014) | 8/1 | 8/16 | 998 | 36.1 |
| | Joaquin (2015) | 9/28 | 10/15 | 931 | 69.4 |
| | Matthew (2016) | 9/27 | 10/7 | 934 | 75 |
| | Jose (2017) | 9/5 | 9/25 | 938 | 69.4 |
| | Maria (2017) | 9/16 | 10/2 | 908 | 77 |
| | Irma (2017) | 8/30 | 9/14 | 914 | 79.16 |
| | Florence (2018) | 8/31 | 9/18 | 937 | 66.6 |
| | Nana (2020) | 9/1 | 9/4 | 994 | 33.3 |
| | Teddy (2020) | 9/12 | 9/24 | 945 | 66.1 |
| | Delta (2020) | 10/4 | 10/12 | 953 | 61.1 |
| | Isaias (2020) | 7/30 | 8/5 | 986 | 41.6 |
| Test Set | Dorian (2019) | 8/24 | 9/7 | 910 | 82.7 |
| | Sandy (2012) | 10/22 | 11/2 | 940 | 51.38 |
| | Igor (2010) | 9/8 | 9/23 | 924 | 69.4 |

## 2.2 Methodology

### 2.2.1 The Long Short-Term Memory Network

Originally developed by Hochreiter and Schmidhuber (1997), the LSTM network belongs to a class of recurrent neural

networks (RNNs). Along with its variants, LSTM has been widely used in forecasting and data reconstruction studies (Kim et
al., 2020; Bethel et al., 2021; Gao et al., 2021; Hu et al., 2021; Jörges et al., 2021). It has also been coupled with other machine
learning tools, neural networks, and numerical models (Choi and Lee, 2018; Ali and Prasad, 2019; Fan et al., 2020; Guan,
2020). LSTMs have an advantage over traditional feed-forward neural networks and other RNNs in that they can selectively
remember patterns in data. This is achieved by a series of forget ($f_t$), input ($i_t$), and output ($o_t$) gates. Data passing through
these gates are processed using the sigmoid function ($\sigma$) and the Hadamard product operator ($\odot$; Yu et al., 2019). Each gate
may be computed as follows:
$$f_t = \sigma\big(W_{xf}x_t + W_{hf}h_{t-1} + b_f\big) \tag{1}$$
$$i_t = \sigma(W_{xi}x_t + W_{hi}h_{t-1} + b_i) \tag{2}$$
$$o_t = \sigma(W_{xo}x_t + W_{ho}h_{t-1} + b_o) \tag{3}$$
$$g_t = tanh\big(W_{xg}x_t + W_{hg}h_{t-1} + b_g\big) \tag{4}$$
$$c_t = f_t \odot c_{t-1} + i_t \odot g_t \tag{5}$$
$$h_t = o_t \odot tanh(c_t) \tag{6}$$
where W is each layer's assigned weight, $x_t$ is the input time step t, b is the bias, c is the cell state, and tanh is a hyperbolic
tangent function.
In sequence, the forget gate is used to delete past information where decisions on which information should be deleted is
defined as the value obtained from estimating the sigmoid following receiving $h_{t-1}$ and $x_t$. The sigmoid function output
ranges from 0 to 1 so that if the value is 0, information of the previous state is completely deleted, and if 1, information is
completely preserved. The input gate saves current information and is processed alongside $h_{t-1}$ and $x_t$ before being applied
to the sigmoid function. The resulting information is then processed with the hyperbolic function and Hadamard product
operator before being sent out of the input gate. The strength and direction of information storage in the current cell is
represented by $i_t$ and $g_t$, which respectively range from 0 to 1, and -1 to 1.

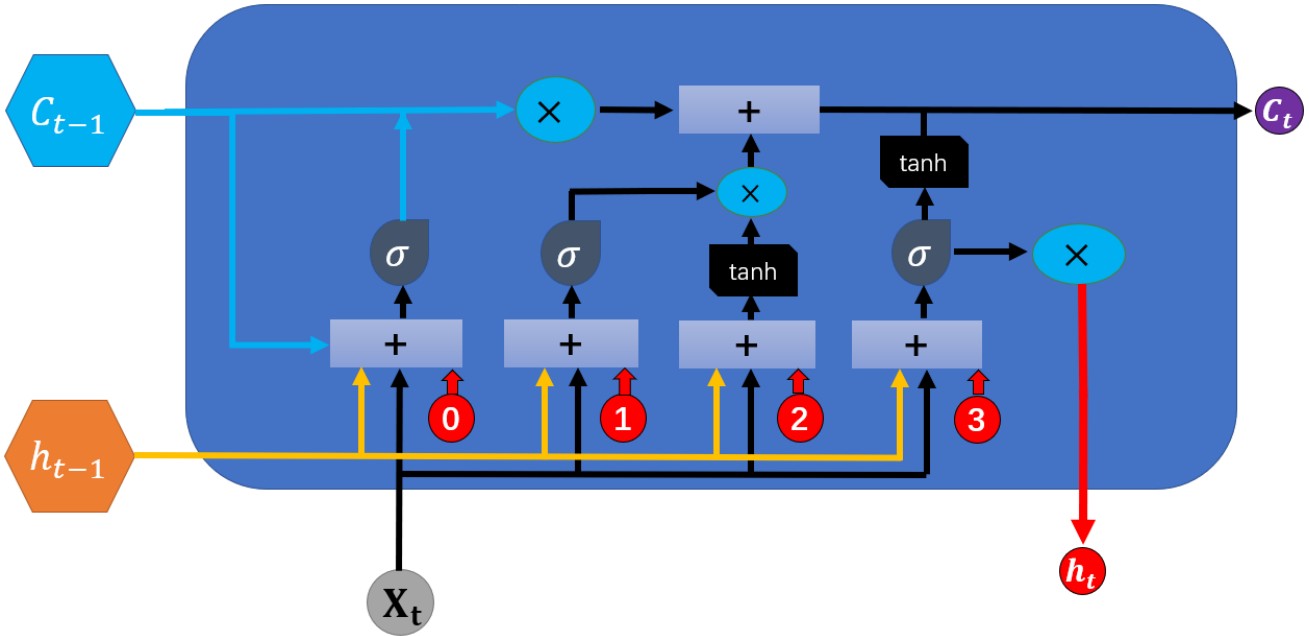

**Figure 2. Architecture of the long short-term memory neural network cell.**

LSTM is set up with four layers that correspond to a time step of four. The recursive linear unit (ReLu) was used as the

activation function to maximize the model's ability to capture nonlinearities. The Adaptive Moment Estimation (Adam)
optimizer is used to compute adaptive learning rates. The number of epochs was set to 100 and the batch size set to 3.
Throughout each experiment, the operating parameters were held constant. These settings were chosen after experiments (not
shown) as they produced the best results while avoiding overfitting. Similar settings can be found in Bethel et al. (2021a) and
Zhou et al. (2021a, 2021b). The data was partitioned along a 70/30 split into training and validation datasets. For clarification,
here, and only here, the word 'dataset' should be interpreted as a given test hurricane (the test set hurricanes of Table 2). A
general model is trained using the training set hurricanes of Table 2, but the model is specified to a given test set hurricane
using 70% of its time series, and the remaining 30% is used to validate the forecast.

**2.2.2**    **Wind Speed Extrapolation**

As seen in Table 1, no buoy measured wind speed at the standard 10 m height and thus, wind speeds were adjusted to this

height using the logarithmic wind profile:
$U_{10} = U_x \frac{\ln(10/Z_0)}{\ln(x/Z_0)}$         (7)
where $U_x$ is the wind speed measured at a given buoy's anemometer height, $x$ is a given buoy's anemometer height, and $Z_0$
is the roughness length (0.0002; Golbazi and Archer, 2019).



### 2.2.3    Performance Indicators

Three commonly used statistical metrics: correlation coefficient (R), root mean square error (RMSE), and mean absolute percentage error (MAPE), are used to assess forecast efficacy. Their equations are as follows:

$$R = 1 - \frac{\sum_{i=1}^{N_i}(x_i - \overline{x_i})(\dot{x}_i - \overline{\dot{x}_i})}{\sqrt{\sum_{i=1}^{N_i}(x_i - \bar{x}_i)^2 \sum_{i=1}^{N_i}(\dot{x}_i - \overline{\dot{x}_i})^2}}$$

$$RMSE = \sqrt{\frac{\sum_{i=1}^{N_i}(x_i - \dot{x}_i)^2}{N_i}} \tag{8}$$

$$MAPE = \frac{1}{N_i}\sum_{i=1}^{N_i}\left|\frac{|x_i - \dot{x}_i|}{x_i}\right| \times 100\%$$

where $x_i$ and $\dot{x}_i$ are the observed and forecasted SWH (m), respectively. $N_i$ is the total number of observations and the overbar denotes averages.

## 3.   Results

### 3.1  Time Series Analysis

To evaluate forecast efficacy, time series of the observed and LSTM-forecasted, hurricane-forced SWHs for Hurricanes Dorian, Sandy, and Igor are given in Figures 3 – 5, respectively. Due to the lack of nearshore buoy observations within The Bahamas, no observations were made when Hurricane Dorian made landfall on Abaco island on September 1st, 2019. NDBC buoy 41010 nevertheless observed the growth of SWH under the influence of the hurricane several hundred kilometres away. In Fig. 3, time series of observed SWH was compared with the nowcast (0-hr, Fig. 3(a) and 3-, 6-, 9-, and 12-hr forecasts (Fig. 3b-e, respectively). In Fig. 3a, it can be observed that an extremely tight fit between the forecasts and observations of Hurricane Dorian-forced SWHs at the start of wave growth from ~3.5 m to just under 7 m. However, at closer inspection, it can also be seen there are periods (e.g., at 42-hrs after UTC 1500 September 1) where the LSTM nowcast is unable to capture the extremely fine details. This is because in addition to errors introduced by LSTM's computations, there are also far too few examples of high-frequency components of the signal that the model could learn from and reproduce. Even following preprocessing using Empirical Mode Decomposition, high-frequency components of original SWH signals remain a challenge for LSTM (Zhou et al., 2021a). Nevertheless, this represents a discrepancy of far less than 1 m and is thus of very little importance when considering estimates of the wave state. When forecasts are performed on a 3-hr horizon, however, discrepancies between observations and the forecast have grown significantly larger where at different times, forecasted SWHs both underestimate and overestimate the observations. This phenomenon is especially noticeable at the 40- and 50-hrs after UTC 1500 September 1 marks. At the 40-hr mark, SWHs were observed by buoy 41010 at approximately 5.5 m, but LSTM predicted a height of only approximately

4.2 m. The difference between the two clearly exceeds 1 m.

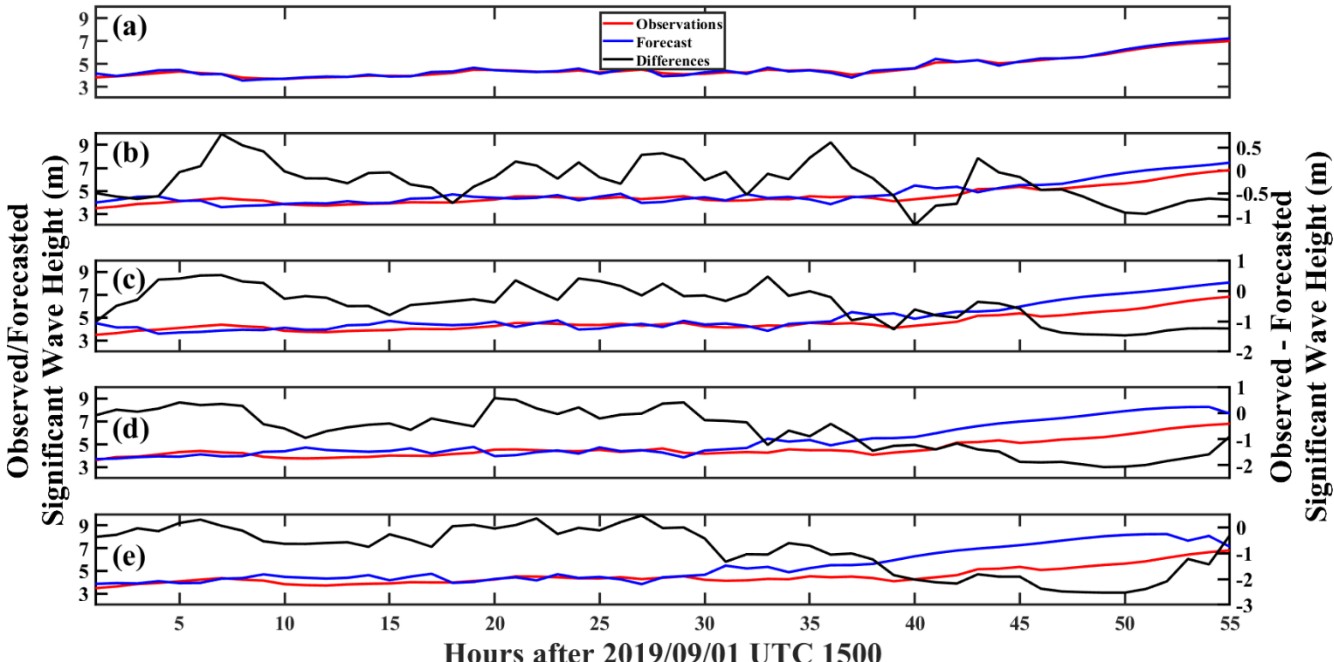


**Figure 3. Time series of Hurricane Dorian observed and LSTM-forecasted SWH (m) at the (a) 0-, (b) 3-, (c) 6-, (d) 9-, and (e) 12-hr**
**horizons, measured at buoy 41010.**
As total wave energy ($P$) is extremely sensitive to SWH (i.e., $P \propto H_s^2 T_p$, where $H_s$ is the SWH and $T_p$ is the wave period),
even minor underestimations of the wave height would lead to radically different energy output. Similarly, at the 50-hr mark,
SWH was measured at approximately 5.6 m, but LSTM forecasted a wave height of approximately 6.5 m. This overestimation
would produce the same radically different energy output than the observations. The same phenomenon can still be observed
for the 6-, 9- and 12-hr forecast horizons respectively presented in Fig. 6c-e, but at a significantly exacerbated scale. In each
case, at the tail end of the forecasts (35+ hrs after UTC 1500 September 1), the distance between the observations and forecasts
widened as the maximum wave height increased.
Identical to Hurricane Dorian, nowcasts of Hurricane Sandy were most efficient at reproducing the observations (Fig. 4a).
Interestingly, though there are some slight differences, LSTM was still able to capture finescale increases or decreases in SWH.
As the forecast horizon is extended to 3-hr in Fig. 4b, however, those finescale details were increasingly missed, though the
general wave growth and decay trends were captured. In Fig. 4c for the 6-hr forecast horizon, and before the 40-hr mark after
UTC 2000 September 10 mark, LSTM nearly consistently underestimated wave heights. Following this point at the peak of
the storm, LSTM virtually captured the observed SWH although finescale details were completely missed. During the wave
height decay stage, LSTM-forecasted wave heights overestimated the observations, but this discrepancy hovered at ~0.5 m and
so, were not as extreme as the discrepancies seen during Hurricane Dorian at the same 6-hr forecast horizon (Fig. 3c). In Fig.
4d and 4e where the 9- and 12-hr forecast horizons are compared with observations, the differences between them is
significantly larger than as compared to the 0-hr nowcast or the 3- and 6- hr forecast horizons of Fig. 4a-c.

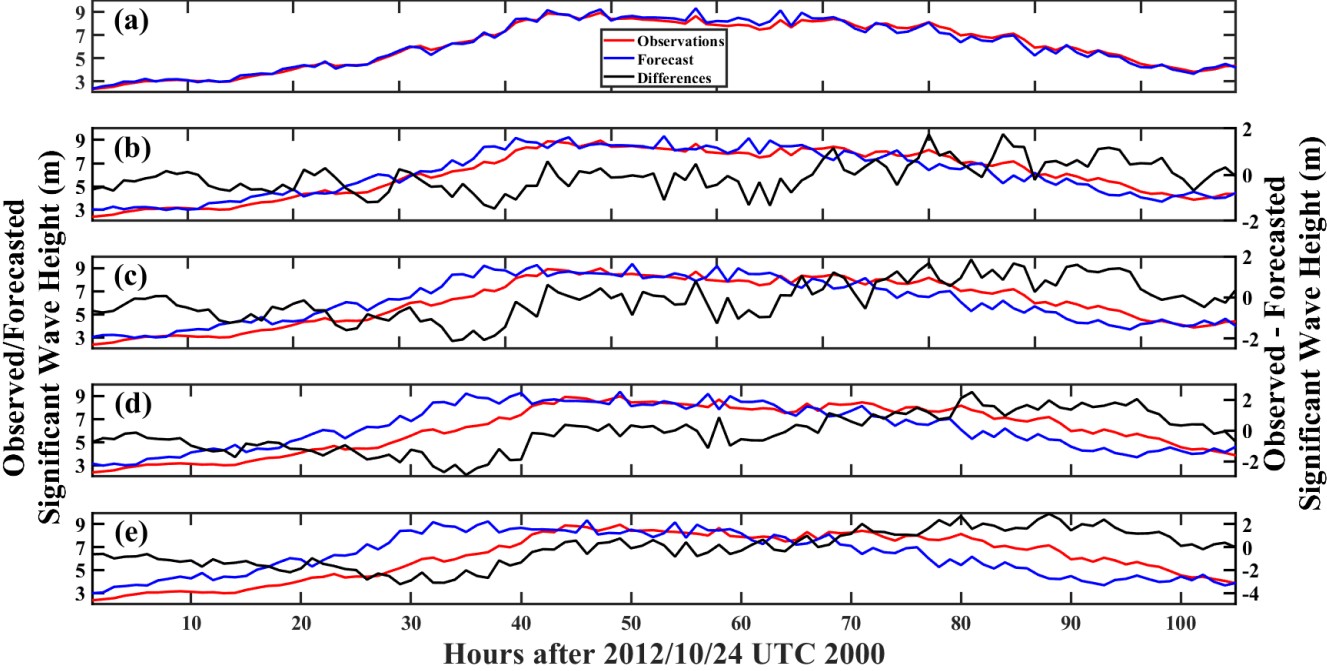

**Figure 4. Same as Figure 3, but for Hurricane Sandy (2012) measured at buoy 42058.**
At its most extreme, the difference between the forecasted (~6 m) and observed (~9 m) SWH reached a staggering 3 m
at the 32-hr mark after UTC 2000 October 24. However, eight hrs later at the peak of the storm, LSTM was once again able to
predict the observed SWHs more adequately. Although LSTM was able to capture the general decreasing, it largely
overestimated the SWH as wave heights began to decrease with the passing of the storm. This overestimation was measured at
approximately 2 m at the 90-hr mark after UTC 2000 October 24.
Although Hurricanes Dorian and Sandy, like Hurricane Igor, were extremely powerful systems, Igor however, spent most
of its time in the Atlantic Ocean far away from any landmasses. Perhaps, then, the maximum wave height was allowed to grow
to just under 11 m as an extremely long, uninterrupted fetch and duration would have been conducive for this wave growth.
This is, of course, tempered by wind energy transfer rates and energy saturation of the wave field (Liu et al., 2008; Hwang and
Fan, 2017; Babanin et al., 2019), in addition to balancing and decay by dissipative forces (Allahdadi et al., 2019; Rollano et
al., 2019; Tamizi et al., 2021). In Fig. 5, similar to the previous two examples, the LSTM nowcast (Fig. 5a) produced
exceptionally accurate results for Hurricane Igor (2010) with regards to the observations.

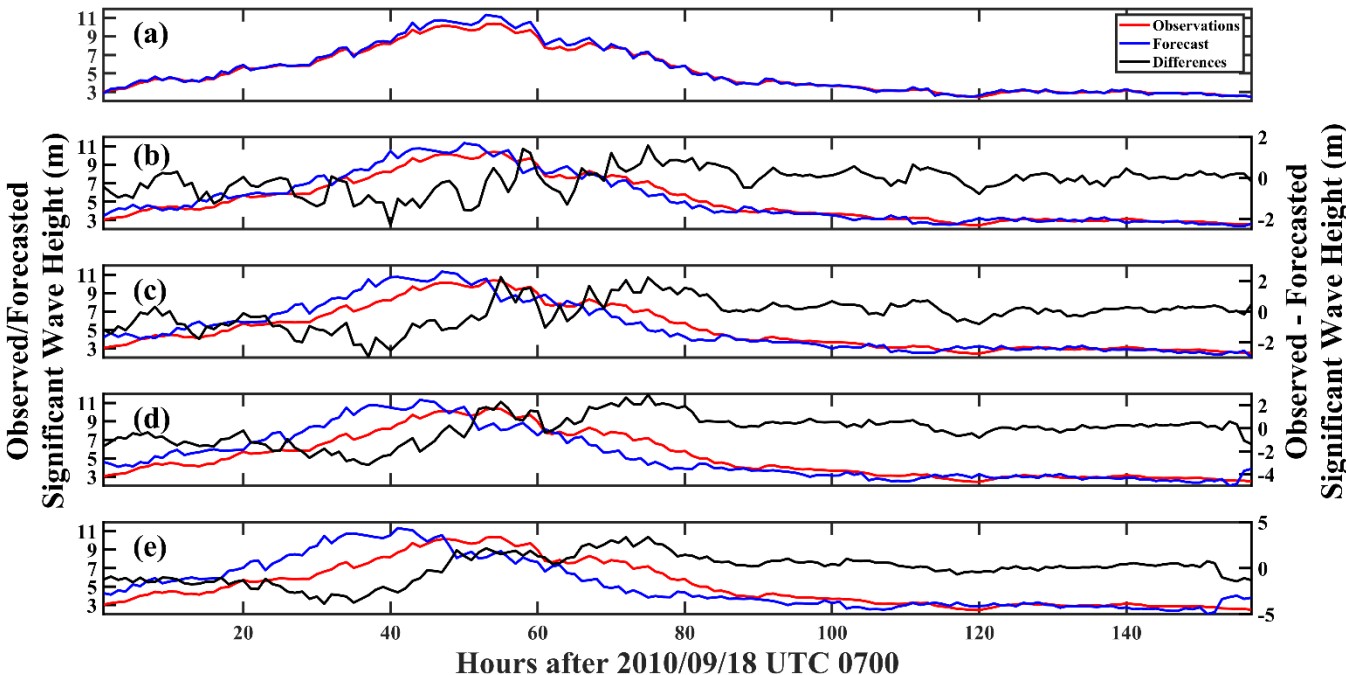

**Figure 5. Same as Figure 3, but for Hurricane Igor (2010) measured at buoys 41048 and 41049.**

This is even true at the peak of the storm at the 50-hr mark after UTC 0700 September 18 when wave heights reached a maximum of just under 10 m. As the forecast horizon increased, however, the same pattern of forecast quality deterioration could be observed where in Fig. 5b at the 3-hr horizon. Although LSTM was able to capture the general trend throughout the time series, LSTM's predictions were slightly out of phase with the observations in its estimation of the point at which the storm generated its maximum wave height (50 hrs after UTC 0700 September 18). This phenomenon becomes increasingly apparent in the 6-hr (Fig. 5c), 9-hr (Fig. 6d) to the 12-hr (Fig. 5e) forecast horizons. Nevertheless, at the tail end of the time series, regardless of the forecast horizon, LSTM produced highly accurate predictions of SWH under forcing by Hurricane Igor (2010).

As the problem is most noticeable here, the problem of LSTM phase shifting during its time series forecasting will be discussed. From Fig. 3, it should be identified that there are lags in forecasts as compared to the observation for Hurricane Igor. This is also observable, but to a much smaller degree in Fig 4. for Hurricane Sandy. Consequently, autocorrelation between time series were estimated and with lag results are presented in Fig. 6. Hurricane Dorian is not shown as its lags were all 0 for each forecast horizon. There, it can be observed that for Hurricane Sandy, the lags increased from 0 hrs at the nowcast (0-hr) and 3-hr forecast, to 1 hr at the 6-hr forecast and continued to increase to 4 hrs at the 12-hr forecast. Similarly, for Hurricane Igor, there was also no lag between the time series from the nowcast (0-hr) and 3-hr forecast, but over time, lags gradually increased from 2 hrs at the 6-hr forecast horizon, to up to 7 hrs at the 12-hr forecast horizon. This occurs because the farther in time predictions are made, errors at each time step builds upon the previous prediction error, thus shifting forecast values.

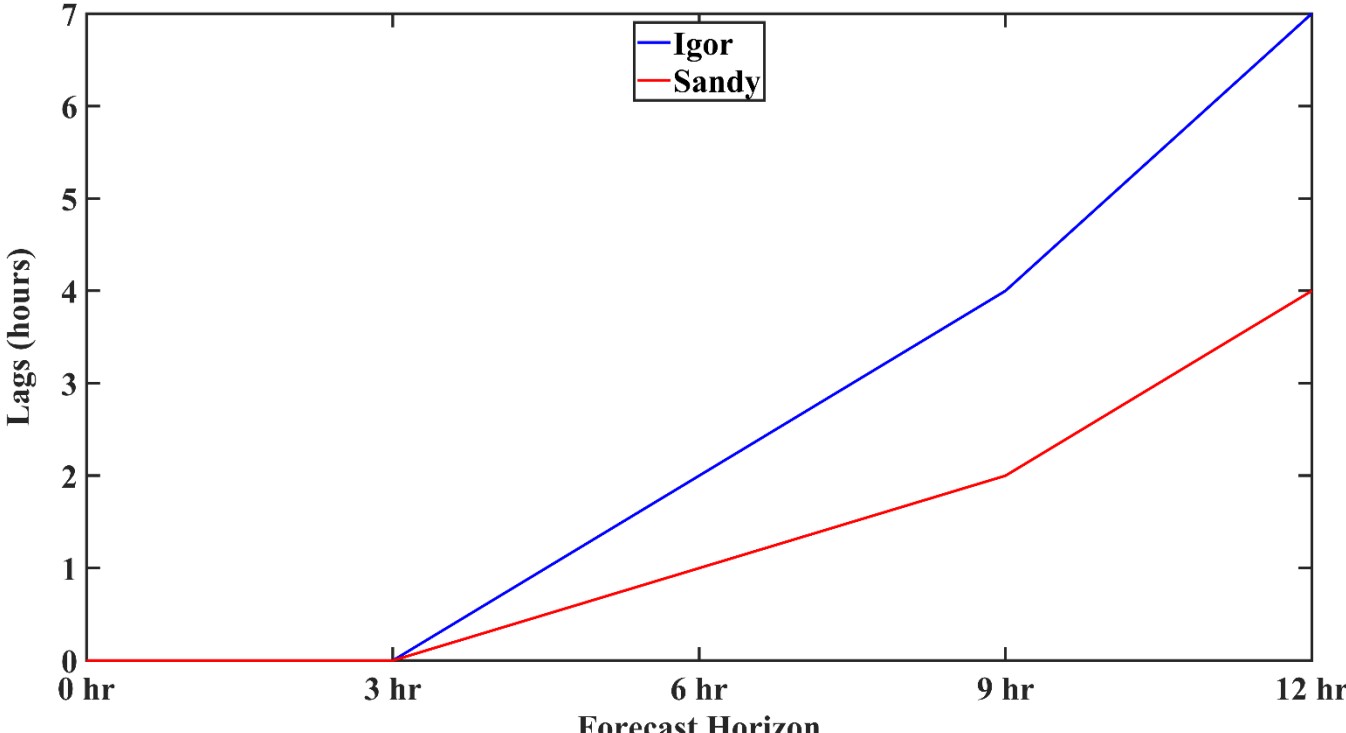

**Figure 6. Estimated lags due to phase shifting of forecasted time series for Igor (blue) and Sandy (red).**
Curiously, the problem of phase shifting and increasing lags over forecast horizon time may also be related to the length
of the time series for a given hurricane event. During experiments, it was noted that as the number of wave height events as
recorded by a buoy during a hurricane increased, the severity of phase shifting also increased alongside observed lags. Data-
driven methods such as LSTM, while they can learn and reproduce the relationships of a variety of climate variables and are
therefore suitable for forecasting, they are prone to making phase shift errors, oscillations, and failures (Kaji et al., 2020l
Morgenstern et al., 2021). Here, Hurricane Igor that possessed the longest time series and as such, its phase shift errors were
most severe, leading to the largest lags between SWH forecast and observation time series. Unfortunately, this and other errors
are inherent to LSTM and may require additional experimentation in modifying the input time series as Morgenstern et al.
(2021) noted that structural changes to LSTM by the usage of encoder/encoder architectures or offsetting the start of forecasts
to the forecast horizon of interest produced no noticeable positive change. While phase shifts and lags represent rather large
disadvantages for this model as it will not be able to accurately predict the timing of, for example, maximum wave heights,
this appears to be only a problem at extended forecast horizons (i.e., 6 hrs and beyond). Nevertheless, the lags are all well
within 12 hrs and thus, although this model should not be depended upon to the exclusion of other forecasting methods, it can
still give several hours of advance warning to coastal communities and regional governments to make minor changes to
hurricane protection plans.

**3.2 Histogram Analysis**
Precise and not merely accurate estimates of hurricane-forced SWHs have the potential to enhance risk assessments and
mitigation strategies as these systems make landfall or approach offshore structures (Hatzikyriakou and Lin, 2017; Marsooli
and Lin, 2018; Masoomi et al., 2018; Guo et al., 2020; Song et al., 2020). This first section investigates the distribution of
forecasted SWHs in comparison with observations for hurricanes Dorian, Sandy, and Igor. In Fig. 7, histograms of observed
and forecasted SWHs under forcing by Hurricane Dorian is presented. In Fig. 7a, it can be observed that for the 0-hr SWH
nowcast, the model approximately exactly matched observations at the 3 – 4 m bin, but minutely underestimated the
observations at the subsequent 4 – 5 m bin. Alternating overestimations and underestimations occurred for the 5 – 6 m and 6 –
7 m bins, but unfortunately, overestimations were most severe at the >8 m bin. There, there were no observed occurrences of
wave heights over 8 m, but the model incorrectly predicted their existence.

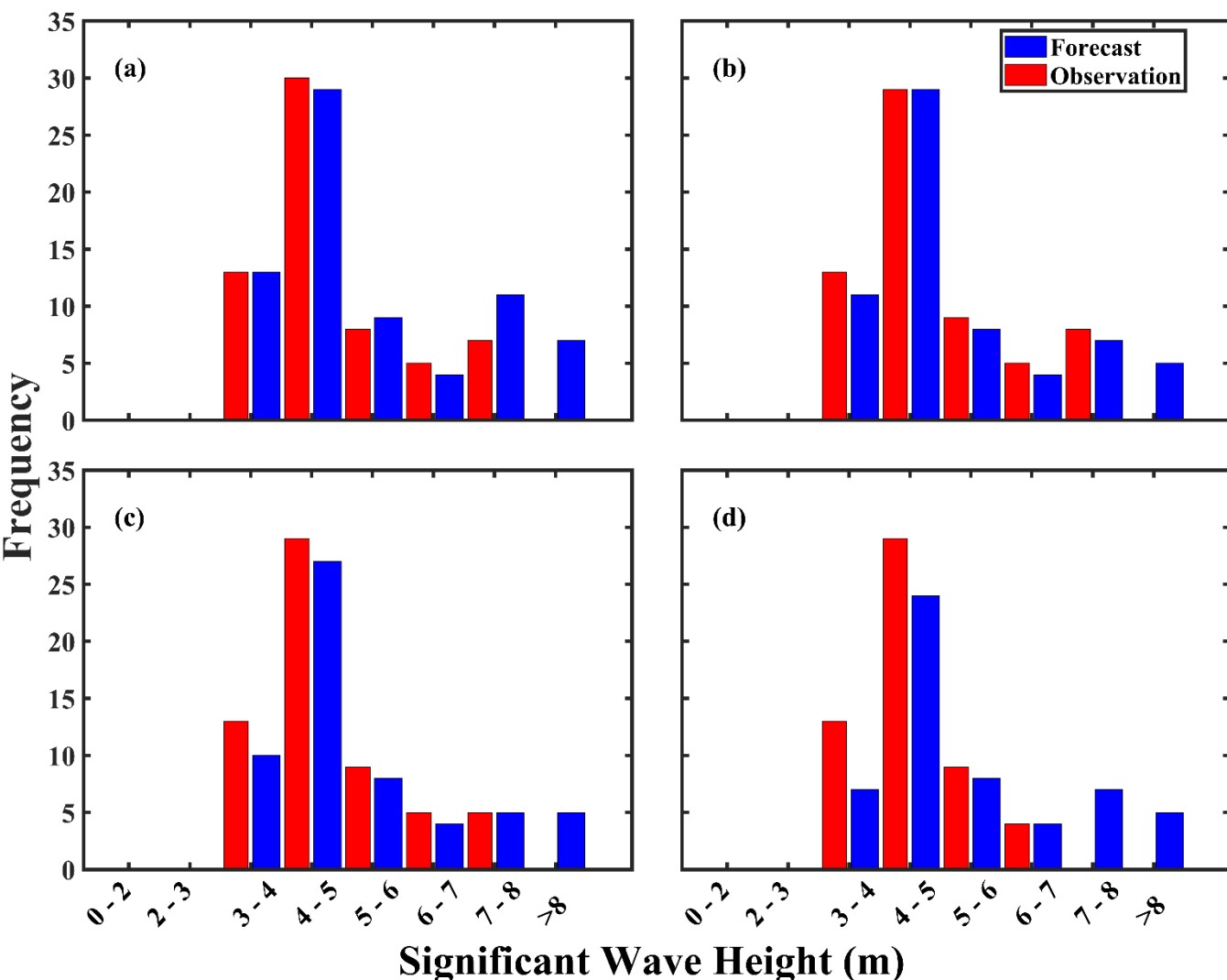


**Figure 7. Histograms of Hurricane Dorian observed (red) vs forecasted (blue) SWH (m) at the (a) 0-, (b) 3-, (c) 6-, and (d) 12-hr**
**forecast horizons. Results for the 9-hr forecast are presented in Figure S1.**
In Fig. 7b, relatively good agreement between the forecasted and observed SWHs, but discrepancies between them have
become increasingly apparent. Though at the 0-hr forecast in Fig. 7a forecasted and observed SWHs exactly matched, LSTM
underestimated the frequency of 3 – 4 m wave heights, but exactly matched the frequency of slightly higher (4 – 5 m) waves.
LSTM underestimations continued through the 6 – 8 m bins, but again, the model overestimated the frequency of waves higher
than 8 m. This trend remains consistent at the 6- and 9-hr forecasts in Fig. 7c and S1, but at the 12-hr forecast in 7d, excluding
the 6 – 7 m and >8 m bins where LSTM respectively exactly matched and overestimated the observations, underestimations of
the frequency of other wave heights occurred at all other bins.
Likewise, Fig 8. presents histograms of observed and nowcasted/forecasted SWHs as forced by Hurricane Sandy. In Fig.
4a, while the maximum wave heights forced by Hurricane Sandy (~9 m) exceeded that of Hurricane Dorian (~8 m), LSTM
was still able to adequately predict the wave height distribution. However, alternating patterns of under- and overestimations
of the frequency of wave heights can still be observed. In Fig. 8a, the 0-hr nowcast underestimated the observations from the
2 – 3 m up to the 4 – 5 m bins before abruptly overestimating all remaining bins, with the >8 m being the most severe case.

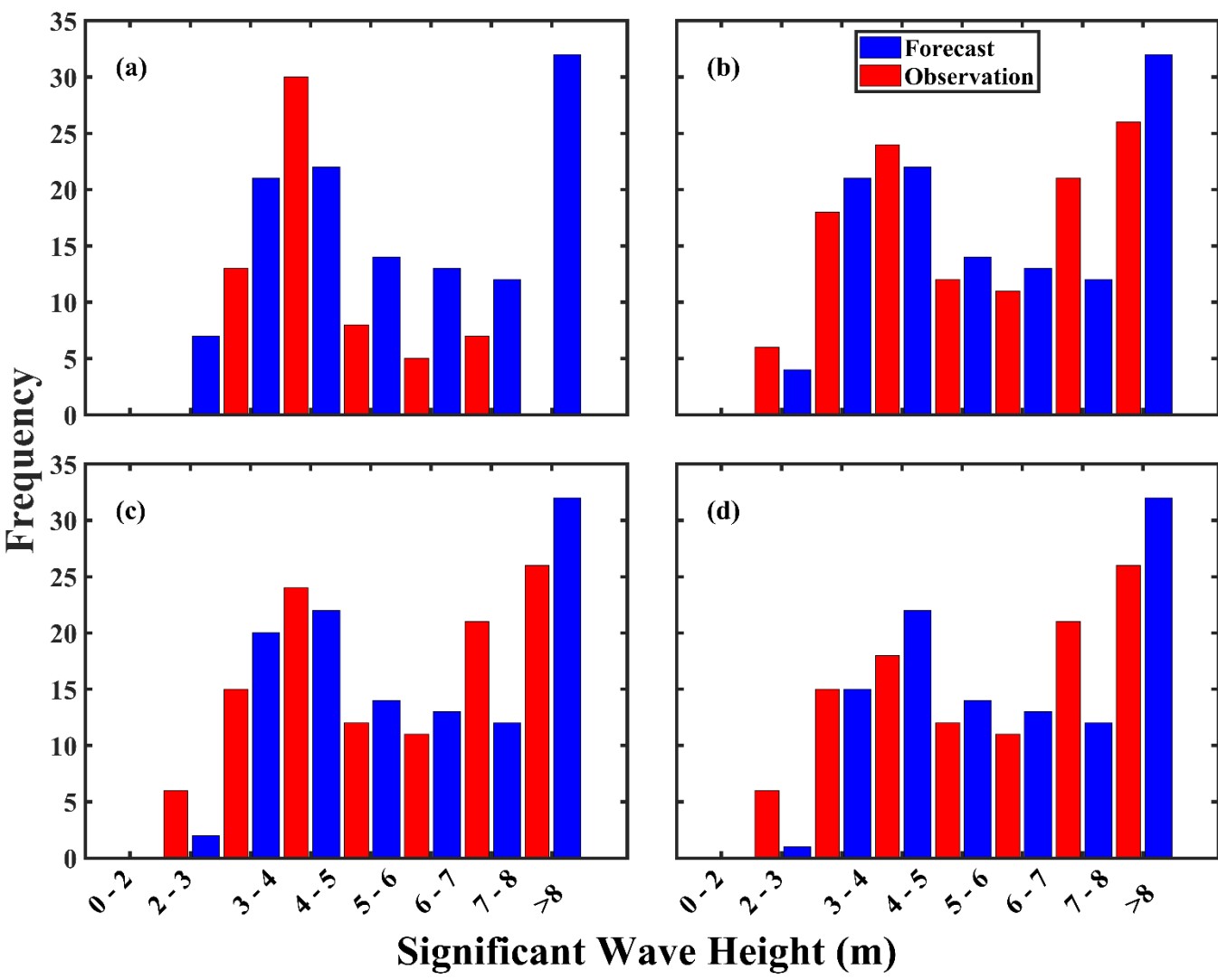


**Figure 8. Same as Figure 7, but for Hurricane Sandy. Results for the 9-hr forecast are presented in Figure S3.**
In Fig 8b at the 3-hr forecast horizon, results are largely improved over the 0-hr nowcast, but underestimations throughout
most of the wave height bins continue. The exception to this remains the overestimation of the frequency of the highest (i.e., >8
m) wave heights. The case remains the same for Figs. 8c, S3, and 8d at the 6-, 9-, and 12-hr forecast horizons.
Results for Hurricane Igor are presented in Fig. 9. Here, Igor produced SWHs that exceeded either Hurricanes Dorian or
Sandy, but interestingly, regardless of the forecast horizon, LSTM was able efficiently (but still imperfectly) forecast the wave
height distribution, even at wave heights up to 9 – 10 m. However, identical to the previous hurricane cases, the frequency of
maximum wave height predictions greater than 10 m is overestimated. Throughout the forecast horizons, naturally, the 0-hr
forecast produced the best results (Fig. 9a). Deterioration of the forecasted wave height frequency and magnitude increased
steadily from the 3-, 6-, 9-, and 12-hr forecast horizons as shown in Fig. 9b-c, S6, and 9d.

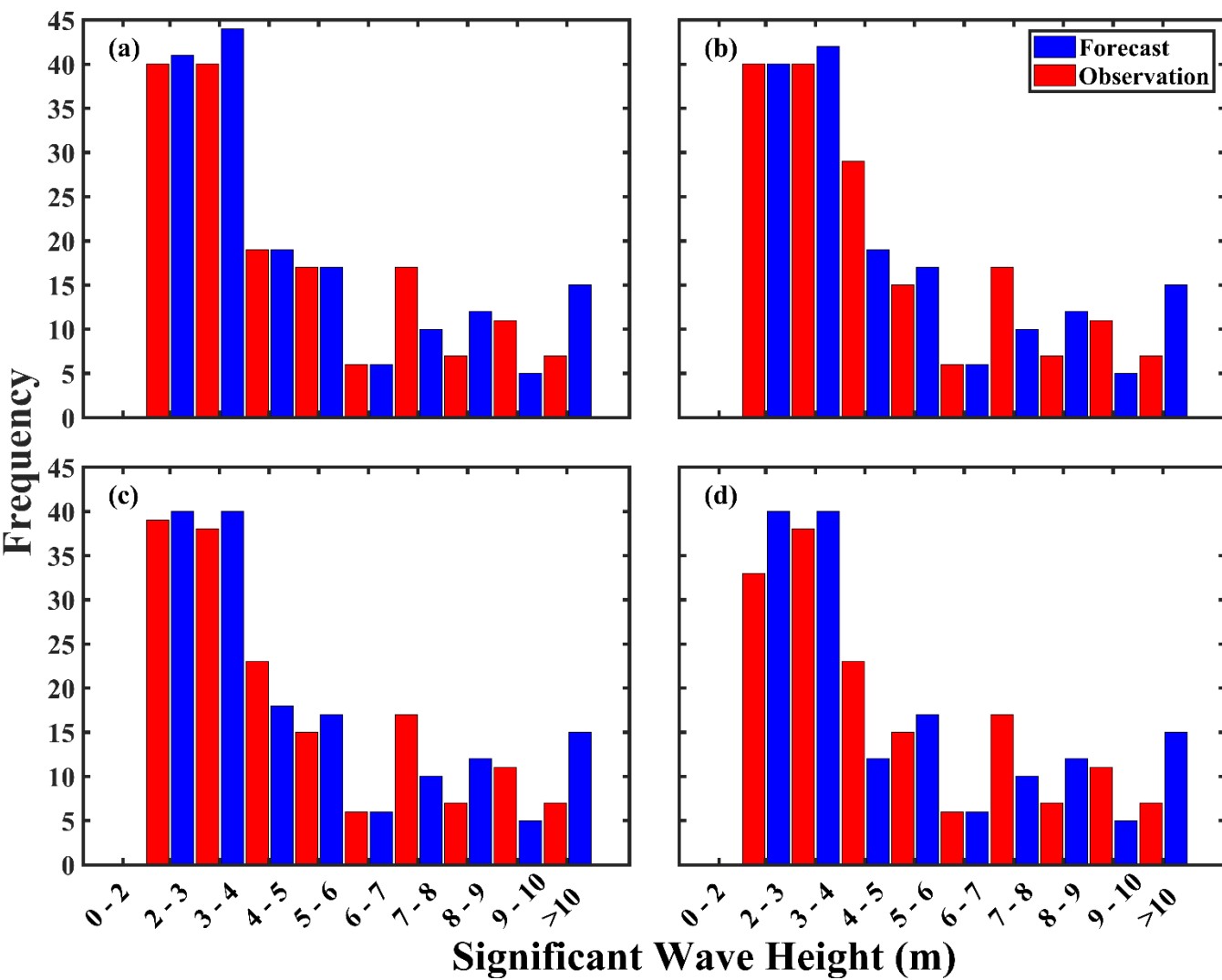


**Figure 9. Same as Figure 7, but for Hurricane Igor (2010). Results for the 9-hr forecast are presented in Figure S4.**
Consistent features of the model are its apparent under- and overestimation of both the frequency of wave heights, and
their magnitudes (Figures S2, S4, and S6). Specifically, the model can underestimate wave heights anywhere by 0.5 – ~2 m in
the cases of Dorian (Figure S2) and Sandy (Figure S6), but also overestimate heights by 2 – 3.5 m. With regards to Igor, this
phenomenon is even more severe with underestimations ranging from 0.5 – ~3 m, and overestimations reaching ~4 m. With
regards to the overestimations, this may indicate that the training dataset contains too many examples of very high wave heights,
which thus necessitates the inclusion of less powerful hurricanes for model training. Though counterintuitive, this is deemed
required as wave growth under hurricane forcing is not merely a function of the maximum wind speed. Indeed, an array of
factors which include, but are certainly not limited to the specific tracks, translation speed and environment (e.g., obstacles
reducing fetch and duration), or modulating factors (e.g., surface currents) all have an impact on wave growth, maintenance,
and decay (Drost et al., 2017; Zhang and Oey, 2018; Hegermiller et al, 2019). Thus, if less powerful hurricanes are considered
in the training dataset as a control (i.e., minimizing the maximum wind speeds available to growth surface waves, regardless
of environment or surface wave-modulating factors), the probability of preferentially populating the training set with large
waves can be decreased. An added benefit would be the inclusion of low wave heights to aid in minimizing underestimation
errors.
**3.3  Total Model Performance**
Overall forecast quality can be assessed through the statistical metrics of R, RMSE, and MAPE, with results for each
hurricane illustrated graphically in Fig. The full range of statistics is available in Table 3. In Fig. 10, it can be observed that
regardless of hurricane, model forecast effectiveness (R) hovered near a perfect 1, but naturally deteriorated over time. By the
3-hr horizon, the three cases diverged from another in reflectance of each hurricane's characteristics. By the 12-hr horizon, the
model was able to maintain accuracies above 0.8 in the majority of cases, which demonstrates that the model remained highly
effective at predictions over a 12-hr time frame. Errors are also minimal: within a 6-hr forecast, RMSEs in all cases can be
maintained under 1 m, but this is increases to just under 1.6 m after a further six hours. Thus, it is suggested that short-range 0
– 6-hr forecasts be prioritized over 12 hours when precision, rather than accuracy is required. Moreover, out of the hurricane
cases, Hurricane Sandy's R performance decreased more rapidly than either Hurricanes Dorian or Igor. This may be related to
the hurricane's track through the central Caribbean Sea (Figure 1). There, both the Caribbean Low-Level Jet (CLLJ) and
Caribbean Current flow in the atmosphere and ocean, respectively.

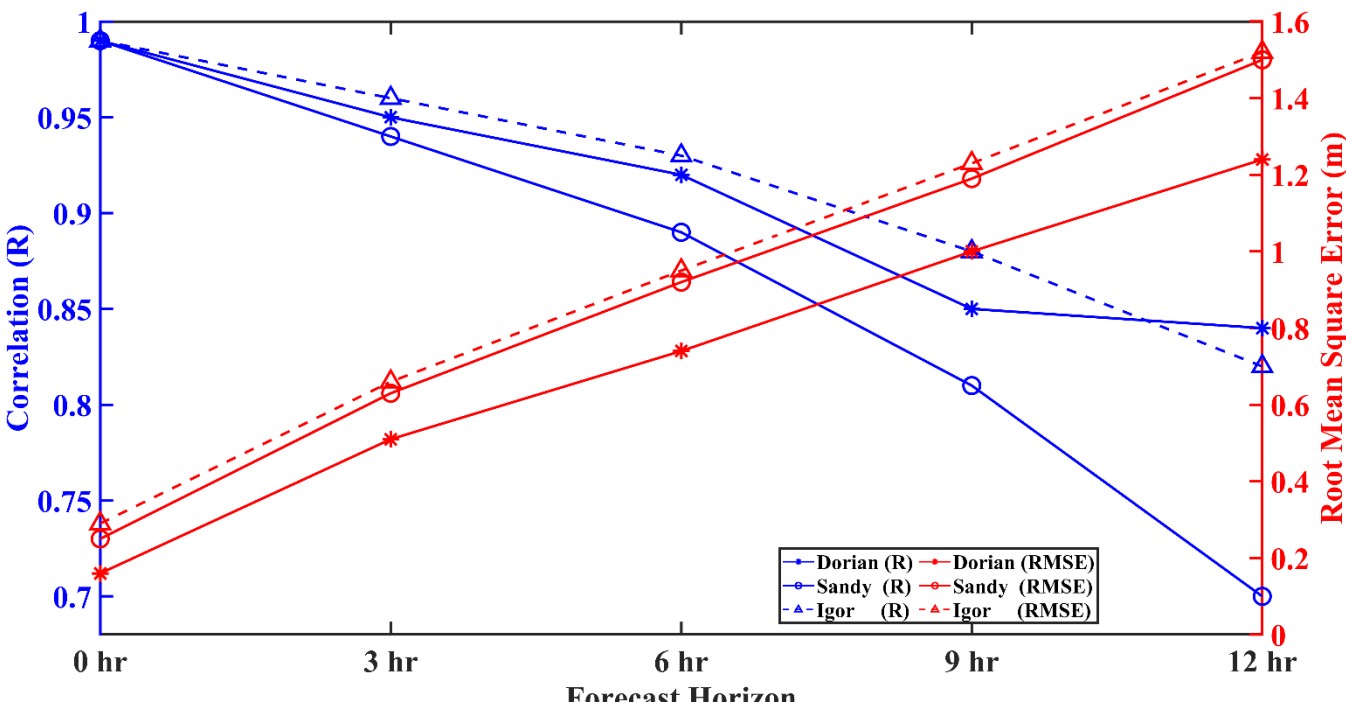

**Figure 10. LSTM model forecast performance in terms of R (blue) and RMSE (red) as compared with the observations for Hurricanes Dorian, Sandy, and Igor.**

It is thought that rather than Sandy's induced wave properties being affected by CLLJ which would have its normal zonal (with the main axis at 15°N) flows disrupted by the hurricane itself, the Caribbean Current would undoubtedly have changed hurricane-induced wave properties. Wave-current interactions have been widely demonstrated to change surface wave properties in a variety of scenarios including, but not limited to tidal flows (Hopkins et al., 2015), large-scale current structures such as the Loop Current and eddies (Romero et al., 2017), but as relevant for this discussion, also hurricane-induced wave interactions with large-scale currents (Sun et al., 2018; Hegermiller, et al., 2019). Unfortunately, as NDBC buoy 42058 that measured the passing of Sandy does not possess surface current information, this hypothesis cannot be tested using the available dataset or possible wave-current effects on hurricane wave field prediction quantified. The rapid decrease in R observed for Sandy could possibly be related to surface current-induced changes in the wave field not accounted for by the dual usage of wind speed and wave height as LSTM predictors for the wave height predictand.

In Fig. 11, the MAPE for each of the hurricanes are given. There, it can be observed that Hurricane Dorian had MAPE values of 2.6% at the 0-hr nowcast and values of 7.99%, 10.83%, 13.13%, and 14.82% respectively at the 3-, 6-, 9-, and 12-hr forecast horizons. By contrast Hurricanes Sandy (Igor) had MAPE values of 3.41% (3.36%), 9.15% (9.53%), 13.34% (13.78%), 17.55% (17.70%), and 22.08% (21.88%) at the 0-, 3-, 6-, 9-, and 12-hr forecast horizons. Both Hurricanes Sandy and Igor had MAPE values approximately 67% higher than that of Hurricane Dorian at the 12-hr horizon.

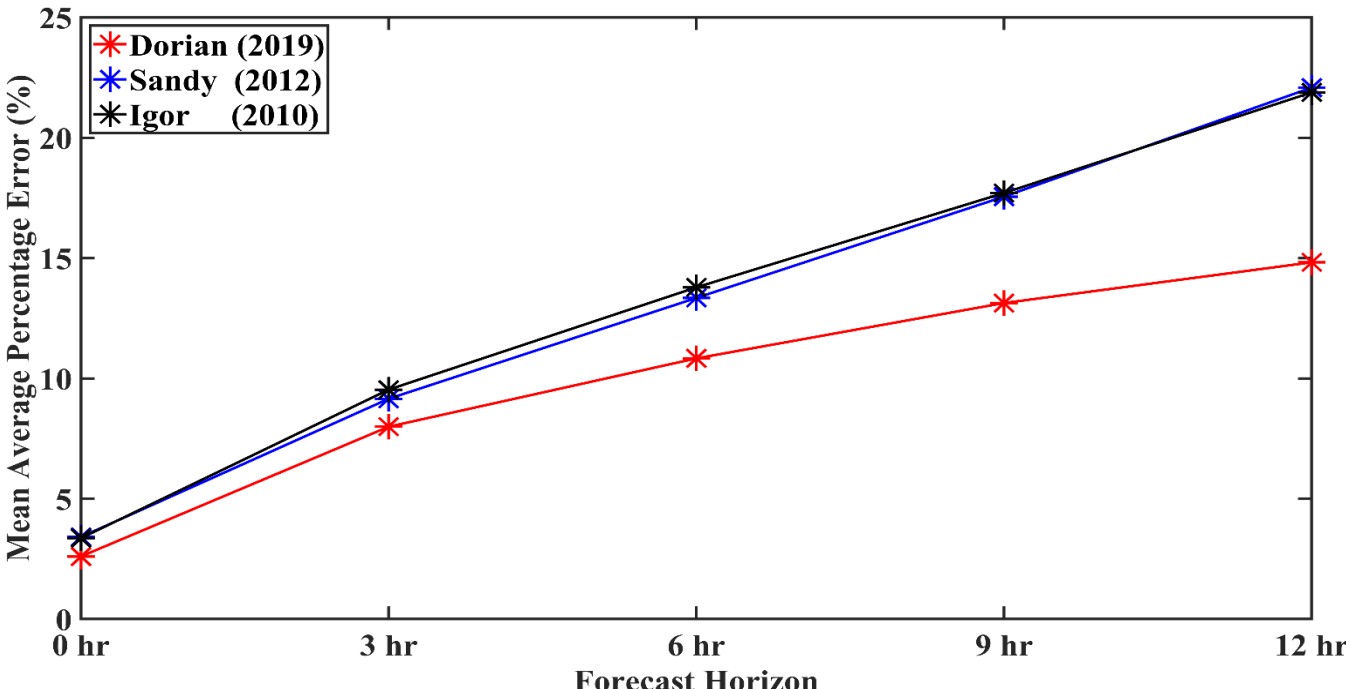

Figure 11. Mean average percentage error (%) for Hurricanes Dorian (red), Sandy (blue), and Igor (black).

The difference in MAPE, in addition to the R and RMSE, may be due to the nature of Hurricane Dorian's time series of wave heights as the system approached NDBC buoy 41010 (Figure 1; Figure 3). Unlike Sandy or Igor where wave heights gradually grew to a peak and then declined, Hurricane Dorian's profile was far more gradual, allowing for LSTM to learn a comparatively much simpler pattern for forecasting. Indeed, unique to Hurricane Dorian, waves induced by the system were only observed after they would have affected and be affected by the Bahamas' continental shelf and its northern islands. As is well understood, islands induce extensive modulation of the oceanic wave field. The presence of islands may cause modifications to wave spectra, reductions in wave heights, and triggering wave diffraction (Cao et al., 2018; Björkqvist et al., 2019; Passaro et al., 2021; Violante-Carvalho et al., 2021). Additionally, as seen for Hurricane Joaquin (2015) by Sahoo et al. (2018), nonlinear wave setup and setdown processes occur when the system interacted with The Bahamas' varying coastal bathymetry, slope, and arching coastlines, and these, in conjunction with Hurricane Dorian's inherent properties (i.e., it's extremely slow translation speed of ~1.4 – 2 m/s), may have all played varying roles in the significantly lower variability in the pattern of wave growth at NDBC buoy 41010.

**Table 3. LSTM forecast performance for Hurricanes Dorian, Sandy, and Igor.**

| | R | | | | | RMSE (m) | | | | | MAPE (%) | | | | |
| | Forecast Hour | | | | | Forecast Hour | | | | | Forecast Hour | | | | |
| | 0 | 3 | 6 | 9 | 12 | 0 | 3 | 6 | 9 | 12 | 0 | 3 | 6 | 9 | 12 |
|---|---|---|---|---|---|---|---|---|---|---|---|---|---|---|---|
| Dorian | 0.99 | 0.95 | 0.92 | 0.85 | 0.84 | 0.16 | 0.51 | 0.74 | 1.00 | 1.24 | 2.6 | 7.99 | 10.83 | 13.13 | 14.82 |
| Sandy | 0.99 | 0.94 | 0.89 | 0.81 | 0.70 | 0.25 | 0.63 | 0.92 | 1.19 | 1.51 | 3.14 | 9.15 | 13.34 | 17.55 | 22.08 |
| Igor | 0.99 | 0.96 | 0.93 | 0.88 | 0.82 | 0.29 | 0.66 | 0.95 | 1.23 | 1.52 | 3.36 | 9.53 | 13.78 | 17.70 | 21.88 |

### 3.4 LSTM Model Comparison

Under the influence of climate change, TCs are widely expected to occur more frequently and with greater ferocity (Chen et al., 2020; Kossin et al., 2020; Geiger et al., 2021). For the CS, the most recent and striking example of this phenomenon occurred during the September 1st, 2019, landfalling of Hurricane Dorian in The Bahamas (Zegarra et al., 2020), which, in addition to damage caused by extremely strong winds and storm surge, hurricane-forced SWHs more than likely added to the damage. Thus, predicting these and other hurricane-forced wave events is of extreme importance, but for Caribbean and other SIDS around the world, these predictions should be of the highest accuracy and where possible, precision, timely, and of minimum required computational expense and expertise (Bethel et al., 2021b). In Figure 12, a comparison is made between the LSTM nowcasted (0-hr) SWH from Figure 3a with SWAN simulations of the same period of time (for model description, see Bethel et al., 2021a), and the observations. Top right and bottom left insets present the position and wind speed of Hurricane Dorian at the start and end of the time series, respectively.

Primarily, the most significant feature in the comparison between SWAN-simulated and LSTM-nowcasted SWHs is that with regards to the observations, LSTM nowcasts are far more accurate at reproducing the time series than SWAN. At the start of the time series (up to ~30 hrs after 1500 UTC September 1st, 2019), the discrepancy between the LSTM nowcast and observations are minimal, while SWAN simulations suggest wave heights of just under 2 m, though observations are just over 3 m. This is remarkable as at that time, the storm was briefly stalled over The Bahamas but waves radiating out could still grow the SWH kilometres away at NDBC buoy 41010 to be recorded. With wind speeds reaching and exceeding 80 m/s, wave heights were just over twice the climatological mean. Following training by past hurricanes, LSTM nowcasts of Hurricane Dorian were very efficient at recreating the observed time series, but at this juncture, SWAN was very notably unable to do so. This may be potentially caused by the usage of low spatial resolution (0.5° × 0.5°) WaveWatch III reanalysis to fill in gaps in buoy data (the 'observations'), thus leading to wide deviations from the SWAN-simulated SWH that possesses a significantly higher spatial resolution (0.2° × 0.2°). This phenomenon, however, should not be used to suggest SWAN simulations are inaccurate. Indeed, after the 30-hr mark following 1500 UTC September 1st, as Hurricane Dorian had migrated away from The Bahamas and decreased in intensity, SWAN's capability at simulating SWHs dramatically increased, just as wave heights began to increase when the system's distance (and maximum wind speeds) from buoy 41010 decreased. Here, though SWAN nevertheless overestimated wave height observations from 30 – 50 hrs after the start of the time series. Again, LSTM did a much better job at recreating the observations but interestingly, after this point, LSTM and SWAN exactly match one another, though they both overestimate the observations. This is a common feature between the data- and physics-

driven approaches at this time and to resolve them, two different approaches are required. Firstly, as previously identified,
the LSTM data-driven approach would require a few more examples of weaker storms to provide lower wave heights in the
training dataset, and this may have a beneficial effect on minimizing overestimations. The physics-based SWAN model, by
contrast, could be improved by advancing model-guiding physics (e.g., Aydoğan and Ayat, 2021), a better representation of
the wind field (Christakos et al., 2020) or online coupling with an atmospheric model such as the Weather Research and
Forecasting (WRF) model (Lim Kam Sian et al., 2020). It should be readily noted at this point that improving physics-based
models require far greater computational resources and expertise than does optimizing training sets for data-driven methods
such as LSTM.

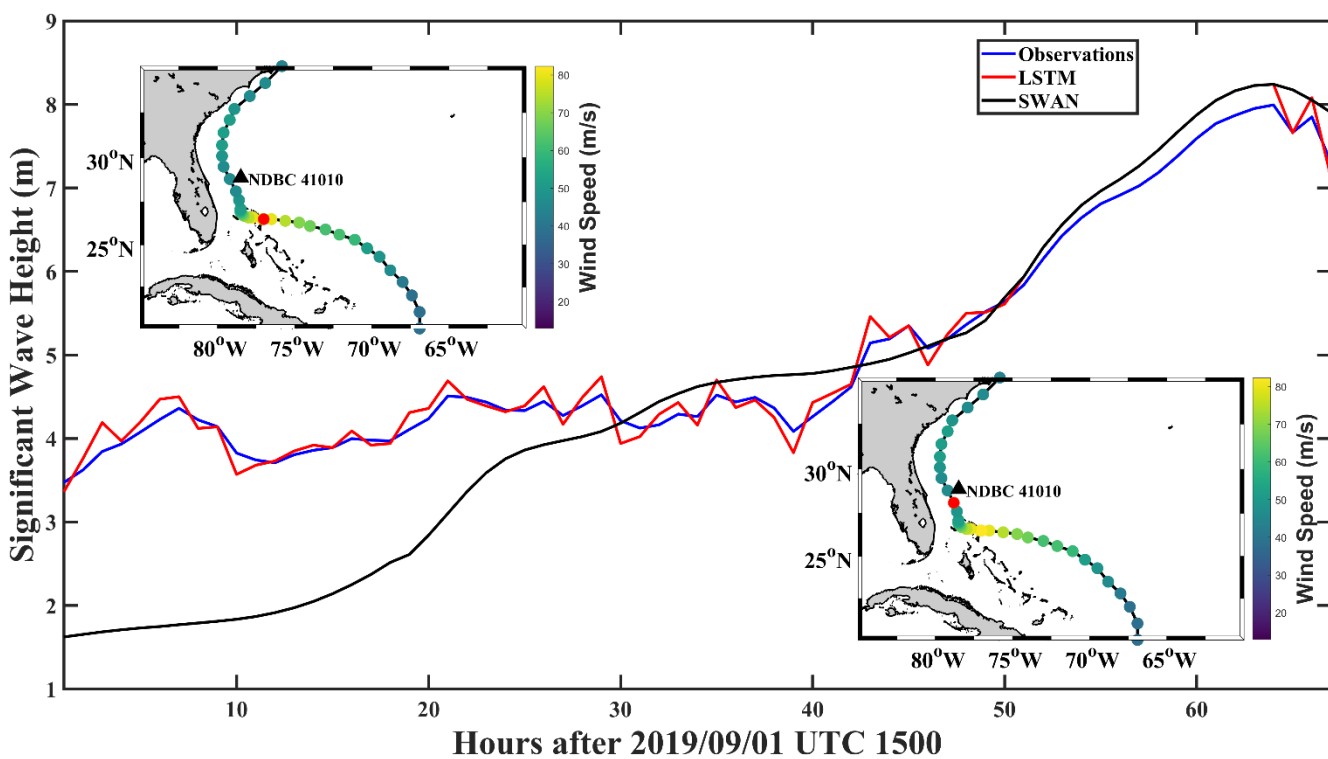


**Figure 12. Comparison of SWH observations (blue), LSTM nowcast (red), and SWAN simulations (black) during (top left inset)**
**and after (bottom right inset) Hurricane Dorian's landfalling in The Bahamas. Red dots indicate the location of Hurricane Dorian**
**in either case.**

Demonstrating, a comparative analysis between LSTM and SWAN for SWH modeling is presented from the

perspectives of required model training/spinup and run times, in addition to their system and expertise requirements (Table
4). There, it can be noted that model training for LSTM took approximately 10 minutes, while for SWAN, model spinup took
just over half an hr. From there, LSTM forecasts took under a second to complete in a personal computer-based Python-
language integrated development environment (PyCharm), while the full run of SWAN took three hrs on two Xeon Gold

6152 CPU processors using a modest 56 cores. The SWAN run must also be understood in the context of the time and expertise needed for preprocessing (i.e., preparing input wind fields, bathymetry, and boundary conditions), in addition to considerations of further modeler skill and experience for processing and postprocess. Though SWAN allows for real-world physics to be considered and thus the model can provide a far greater array of variables to a high degree of accuracy with regards to observations, the CS and other SIDS around the world largely do not have either the required computational resources or human resources to use these and other numerical models. Data-driven methods such as LSTM should therefore be used to supplement existing forecasting tools considering their ease of use, accuracy, and low expertise and computational resource requirements.

This study presented a 1D case, but the work here is easily extended to a 2D case as shown by Zhou et al. (2021b). There, a ConvLSTM model was used on a GeForce RTX 2080 Ti graphics card for hurricane-forced SWH training and forecasting. Very high accuracies with regards to a WaveWatch III baseline was achieved. Crucially, ConvLSTM model training took only 2 hrs and forecasting took just under 20 seconds, which easily outperforms SWAN (here) in terms of speed, and thus could be a viable alternative to the pure usage of numerical wave models under both mean and extreme (i.e., TC-forced) wave conditions.

**Table 4. Model comparative analysis.**

| Model | Training/Spinup Time (hr) | Model Run Time (hr) | Utilized Processor | Expertise Requirements |
|---|---|---|---|---|
| LSTM | 1/6 | $\ll 1/60$ | Intel Core i7-10510U | Minor |
| SWAN | 1/2 | 3 | Xeon Gold 6152 CPU | Major |

**4. Discussion**

Forecasting hurricane activity and its properties remains a daunting task for the scientific community, but great strides have been made in the development of statistical/probabilistic methods, numerical models, and as presented in this study, AI techniques. The results of this study are in strong agreement with those observed by Meng et al. (2021) and Wei (2021) that each found that AI was highly effective at predicting hurricane-induced SWHs. However, although contemporary applications of AI in the forecasting of both in mean and extreme (i.e., TC-forced) waves states have relied traditionally on singular inputs of SWH (Ali and Prasad, 2019; Zhao and Wang, 2018; Zhou et al., 2021a, b), a growing body of literature have demonstrated that the addition of other variables such as wind speed (as done here), wind direction and other variables improves forecast

effectiveness (Kaloop et al., 2020; Zubier, 2020; Raj and Brown, 2021; Wang et al., 2021). Uncertainties in variable selection
have also stimulated research into how to best identify predictors for the SWH or other predictands (Li and Liu, 2020; Li et
al., 2021). These results nevertheless remain consistent with the findings of Chen and Wang (2020) where the introduction
of meteorological data could improve wave forecasts, but longer forecast horizons led to underestimations of extreme wave
heights.
Moreover, discrepancies in forecasting outcomes between hurricanes in this study are slight, but noticeable. This may
reflect differences in LSTM training and test hurricane properties. These include hurricane wind field, translation speed,
approach angle and track which have been demonstrated to be essential factors in governing wave evolution (Zhang and
Oey, 2018; Zhang and Li, 2019; Wang et al., 2020). For example, as a hurricane translated through the study area, wave
properties in any of the four quadrants could have been measured by the chance intersection of the hurricane and its
observing buoy (Zhang and Oey, 2018; Tamizi and Young, 2020; Tian et al., 2020; Collins et al., 2021). Thus, the model
may have learned too much information from a particular quadrant. Consequently, when encountering a different
quadrant in a forecasted hurricane, its results would naturally be poorer than if the model was trained solely on SWHs
from quadrant A in training sets and forecasted quadrant A in the test set. Further experimentation would be required to
identify the difference, if any, and magnitude of using data from a particular quadrant in a hurricane in the prediction of
a different quadrant in a future hurricane. Other variables to consider, especially in the case of those hurricanes in the
CS given its numerous islands, are the morphology of those islands as they can have a strong influence on local ocean
dynamics (Cheriton et al., 2021). For those hurricanes that made landfall in The Bahamas, additional consideration
should be given to the nonlinear interactions that hurricane waves and storm surge have on the archipelago's narrow
and steep carbonate shelf and its variability due to elongated coastlines (Sahoo et al., 2019). These can perhaps be dealt
with by the special application of a combination of a high order spectral method with Krylov subspace techniques as
pioneered by Köllisch et al. (2018). Another set of examples come from Puerto Rico and the U.S. Virgin Islands (Joyce
et al., 2019), and the shallow continental shelf between India and Sri Lanka (Sahoo et al., 2021). Consequently, training
and test datasets certainly contain data from any of a hurricane's four quadrants, or in the case of Hurricanes Joaquin
(2015) and Dorian data recorded along The Bahamas' vulnerable, eastern-most, Atlantic Ocean-facing islands. In these
terms, the effect of training data selection on overall forecast quality has yet to be quantified and should be assessed.
Following this, finescale LSTM-based hurricane-forced SWH forecast models for a given CS country or territory could
potentially benefit from increased discrimination in selecting hurricane training data.

Accompanying increased scrutiny in building LSTM training datasets to improve predictions, the usage of physics-

based/informed/infused versions of LSTM and other artificial intelligence and machine learning algorithms (Karniadakis et
al., 2021; Zhang et al., 2021) may help to bridge the gap in forecasting efficacy between physics-based third-generation
numerical wave models such as WaveWatch III or SWAN. Crucially, this will ensure that forecasting remains significantly
computationally cheaper than the sole usage of wave models. These methods have been successfully applied to the solving
of differential equations in engineering (Niaki et al., 2021; Zobeiry, and Humfeld, 2021), analyzing blood flow (Arzani et al.,
2021), and chaotic systems (Khodkar and Hassanzadeh, 2021). Relevant for the current discussion, these methods are also
finding use in weather and climate modelling (Kashinath et al., 2021). Considering the large physical complexities in wave
evolution under TC forcing (Tamizi et al., 2021), and the many nonlinearities that govern crucial processes (Yim et al., 2017;
Constantin, 2018; Sharifineyestani and Tahvildari, 2021), incorporating physics-informed, or knowledge-guided machine
learning should, respectively, improve and lengthen forecast efficacy and horizons.
**5.  Conclusion**

Precise, computationally cheap, and rapid SWH forecasting under hurricane forcing is of immense value to safeguard

lives, property, and economic development in coastal communities and especially, SIDS. This study used surface wind speed
and SWH forced by 17 hurricanes as input to the LSTM neural network to nowcast and forecast SWHs in the CS. Three
hurricanes, Dorian (2019), Sandy (2012), and Igor (2010) were used as test cases. Results illustrated that the model was
highly accurate at reproducing observed hurricane-forced wave height distributions both in terms of magnitude and frequency.
However, there were discrepancies between observations and predictions. This was most easily observable from the
comparison of observed and forecasted SWH time series for the three test cases.

In all cases, although the nowcasts naturally produced the best results, instances of slight under- and overestimations

could nevertheless be observed at many finescale details. These under- and overestimations became more severe with
increasing forecast horizon length. It has been demonstrated that wave height nowcasting (i.e., a forecast horizon of 0-hr)
was very effective where in the test cases of Hurricanes Dorian (2019), Sandy (2012), and Igor (2010), R (RMSE) was
measured at 0.99 (0.16 m), 0.99 (0.25 m), and 0.99 (0.29 m), respectively. Corresponding values of MAPE for Dorian, Sandy,
and Igor were measured at 2.6%, 3.14%, and 3.36%, respectively. For forecast horizons ranging from 3-, 6-, 9-, and 12-hrs,
with regards to observations, Dorian predictions produced R (RMSE; MAPE) values of 0.95 (0.51 m; 7.99%), 0.92 (0.74 m;
10.83%), 0.85 (1 m; 13.13%) and 0.84 (1.24 m; 14.82%), respectively. Similarly, with regards to observations, Sandy
predictions produced R (RMSE; MAPE) values of 0.94 (0.63 m; 9.15%), 0.89 (0.92 m; 13.34%), 0.81 (1.19 m; 17.55%) and
0.70 (1.51 m; 22.08%), respectively. Igor predictions produced R (RMSE; MAPE) values of 0.96 (0.66 m; 9.53%), 0.93 (0.95
m; 13.78%), 0.88 (1.23 m; 17.70%) and 0.82 (1.52 m; 21.88%), respectively. In general, the model can provide forecasts with
errors of 1 m within 6 hrs of lead time, and an accuracy of greater than 80% up to 12 hrs.

LSTM forecasts were also compared with a widely-used third generation model, SWAN in terms of model accuracy,

computational expense, and difficulty of usage. Using Hurricane Dorian as an example, the data-driven LSTM model was,
over the short-range nowcast, were far more accurate than SWAN. This is a trend widely observed in the literature (see
Reikard and Rogers, 2011 for an excellent treatment on the subject). SWAN nevertheless was capable of simulating observed
SWHs at the peak of the storm and here, achieved parity with LSTM for a brief period of time, demonstrating that within
narrow windows, LSTM can provide accurate estimations of hurricane-forced wave fields, but crucially at a much faster pace
and cheaper computational costs. Despite this, the study is limited in four significant ways.

Firstly, identical to Meng et al. (2021), this study focused on forecasting hurricane-forced SWHs, rather than mean states.

Although a large number of hurricanes occurred over the study period, only a minority of these hurricanes were observed by
buoys. Thus, the LSTM training datasets were severely limited in hurricane cases. This would have a significant effect on
reducing forecast horizons and overall forecasting efficacy. A significantly expanded array of observational platforms in the
Caribbean (i.e., both in situ buoys and remote sensing high-frequency coastal radars) would increase the likelihood of crucial
hurricane wind/wave properties being observed in sufficiently high resolutions to make future research such as this possible.
Secondly, and perhaps more importantly, as TCs and their properties rapidly evolve in space and time (Leroux et al., 2018;
Bhalachandran et al., 2019; Chen et al., 2021), they naturally have great implications on the properties of waves they excite
(Haryanto et al., 2021). If these properties change rapidly enough, LSTM alone would be unable to capture their
characteristics. A recent study by Zhou et al. (2021b) demonstrated that an integrated EMD-LSTM model is more effective
at forecasting rapidly evolving and large wave heights, but whether this remains true for hurricane-forced waves remains to
be seen. Future research should investigate the efficacy of the EMD-LSTM model in forecasting hurricane-forced wave
heights, and a ConvLSTM model fed with high-resolution wave data should be employed for two-dimensional hurricane-
forced SWH. Thirdly, the selection of training and test sets would have an extremely strong impact on forecasting results.
Specifically, Hurricanes Dorian, Sandy, and Igor were are all far more powerful than hurricanes within the training set. These
were chosen as it is expected that due to climate change, hurricanes are due to not only become more frequent, but also, more
intense. The present method demonstrates that the model overestimates the highest SWHs of even those systems and should
continue be effective if hurricanes become even more extreme (and thus, the degree by which the current model overestimates

maximum SWHs should decrease). However, if future systems are weaker than the test set (as it is now), the problem of overestimation would be exacerbated. Thus, a second model that is trained with hurricanes even weaker than the training set would be prudent and run in parallel to ensure both scenarios are considered in future disaster aversion strategies. Fourthly, LSTM-phase shifting of forecasted time series and resultant lags, seen most notably in Hurricanes Sandy and Igor, is a problem that needs to be rectified before the model can be used in real-world, operational TC wave forecasting applications. Extensive research into the mathematical principles underlying LSTM should be conducted by SIDS in the CS and around the world to realize low-cost but high-accuracy forecasts.

**Data Availability:** Buoy datasets are provided by the National Data Buoy Center and can be accessed at https://www.ndbc.noaa.gov/. Hurricane statistics can be acquired from the National Hurricane Center at https://www.nhc.noaa.gov/. WaveWatch III reanalysis data as provided by the Pacific Islands Observing System can be acquired at https://coastwatch.pfeg.noaa.gov/.

**Author Contributions:** BJB, WJS, CD and DXW designed the experiments and BJB carried them out. BJB developed the model code and performed the simulations. BJB prepared the manuscript with contributions from all co-authors.

**Acknowledgements:** The National Data Buoy Center is greatly thanked for the continued maintenance of its buoy array in the Caribbean and for ensuring the public accessibility of its data. The National Hurricane Center is thanked for providing the hurricane statistics and the Pacific Islands Ocean Observing System is thanked for providing WaveWatch III reanalysis data.

**Funding:** This work was supported by the Southern Marine Science and Engineering Guangdong Laboratory (Zhuhai) (SML2020SP007), the Innovation Group Project of the Southern Marine Science and Engineering Guangdong (Zhuhai) under contract No. 311020004, and the National Key Research and Development Program of China (2017YFA0604100, 2016YFC1402004 and 2017YFC1404200).

**Competing Interests:** The authors declare that they have no conflict of interest.

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
