# Peer review of "Forecasting Hurricane-forced Significant Wave Heights using the Long Short-Term Memory Network in the Caribbean Sea"

_Ocean Science, 2021_

## Author Response (AR1)

Reviewer 1 Comments

This paper presents an interesting application of a recurrent neural network for nowcasting and forecasting of significant wave heights in the Caribbean Sea. The authors also provide a useful overview of related machine learning techniques in this field. The method appears to provide useful forecasts at up 12-hour lead times, although while maximum SWHs are reasonably well forecasted, the timing of peak SWH seems to be poorly predicted. However, there are several statements which do not appear to be supported by the current figures and there are several issues with the manuscript which require clarification:

Reviewer Comment 1: Representativeness of the training set. As noted by the authors, due to the focus on extreme conditions, the training set is relatively limited. However, I think the authors need to discuss further the expected effects of this limited training set.

Response: While we would like to go into further detail on the expected effects of the limited training set, we have provided what we think could happen in Line 430 of the updated manuscript, but we cannot further expound upon it as we have not yet run the experiments, nor is there literature on this particular problem as this is a very, very new field.

"Thus, the model may have learned too much information from a particular quadrant and thus when encountering a different quadrant in a forecasted hurricane, its results would naturally be poorer than if the model was trained solely on SWHs from quadrant A in training sets and forecasted quadrant A in the test set. Further experimentation would be required to identify the difference, if any, and magnitude of using data from a particular quadrant in a hurricane in the prediction of a different quadrant in a future hurricane."

Reviewer Comment 2: Additionally, from Table 2 the test data have generally deeper lows and faster wind speeds than the training set. Some discussion on the use of this method on hurricanes which fall at the extremes, or outside the range of conditions in the training set would be beneficial.

Response: We agree and have added the following discussion at Line 492:

"Thirdly, the selection of training and test sets would have an extremely strong impact on forecasting results. Specifically, test sets of this study (i.e., Hurricanes Dorian, Sandy, and Igor) were are all far more powerful than hurricanes within the training set, and these were chosen as it is expected that due to climate change, hurricanes are due to not only become more frequent, but also, more intense. The present method demonstrates that the model overestimates the highest SWHs of even those systems and should continue be effective if hurricanes become even more extreme (and thus, the degree by which the current model overestimates maximum SWHs should decrease). However, if future systems are weaker than the test set (as it is now), the problem of overestimation would be exacerbated. Thus, a second model that is trained with hurricanes even weaker than the training set would be prudent and run in parallel to ensure both scenarios are considered in future disaster aversion strategies."

Reviewer Comment 3: Line 77: Hurricane Humberto is explicitly excluded from the training set. However, the reasons for this removal are not fully explained: would swell contamination be known a priori to know that any forecast of this hurricane would be unreliable? How would such cases affect the use of this technique in real-time forecasting?

Response: In producing the training set, we noticed that the inclusion of Humberto consistently led to poor results, but once it was excluded, results improved tremendously. To demonstrate, a recent publication of ours (Bethel et al., 2021) shows that as wind speeds in Hurricane Humberto decreased, its wave heights continued to increase, leading to extremely poor results when Caribbean Sea Empirical Wind-Wave Model developed there was applied. Thus, we have excluded it from the present study and as we would potentially not know a priori that swell contamination of the wave field would reduce the efficacy forecasts for real-time forecasting and as such, we have amended that paragraph to go into further discussion:

"In some cases (e.g., Earl (2010), Igor (2010), Dorian (2019), Delta (2020)), the same hurricane was observed multiple times along its track. To increase the total length of

the LSTM training/test sets, these data segments were arranged into a single time series. Additionally, cases such as Hurricane Humberto (2019) were explicitly excluded as swell contamination of the wave field could potentially lead to poor forecasts, despite its classification as a major hurricane, large effects on the marine environment (Avila-Alonso et al., 2021), and damage to the British overseas territory of Bermuda. Indeed, when a recently developed empirical wind-wave model for the CS was applied to Hurricane Humberto by Bethel et al. (2021b), observations of wind speed was a very poor predictor of the wave height and thus, given that surface wind speed and SWH are being used jointly here, worsening of LSTM predictions using Hurricane Humberto in the training dataset is natural. Unfortunately, it may not be possible to know a priori the existence of swell that may interfere with linear wind-wave relationships and as thus, is a disadvantage of the current model."

Bethel, B.J., Dong, C., Wang, J: An Empirical Wind-Wave Model for Hurricane-forced Wind Waves in the Caribbean Sea, Earth Space Sci, 8(12), e2021EA001956, https://doi.org/10.1029/2021EA001956, 2021b.

Reviewer Comment 4: Difficulty in interpretation of figures and confusion of the colour used for observations and forecasts in the text and figure captions. I have found it difficult to interpret Figures 3, 4, and 5. Neither the red nor blue lines in the figures have consistent values between all panels. Why do the observations at the same timesteps change value between the panels?

Response: We apologize for this rather large oversight. The colors used are opposite to what they should be and have corrected all of the figures so that they are in line with the text.

Reviewer Comment 5: Line 133: It is not explicitly stated, but I assume that the forecasts are all of SWH at a point (the buoy locations which provide the validating observations). If not, this should be clarified. Also, an explanation of how and why the nowcast differs from the observations should be given. Are observations up to the current time used in the forecast?

Response: You are correct in your assumption that the forecasts are all of SWH at a point. There are differences between the observations and nowcasts because LSTM

introduces errors in its calculations. We have added that sentence to the manuscript. Also, the observations are indeed up to the current time used in the forecast.

Reviewer Comment 6: Line 151: Referring to Fig3b, the stated observation and forecast values are the opposite of what is shown in the figure.

Response: This was a plotting issue and was corrected.

Reviewer Comment 7: Figure 5: The caption states that the observations are as measured at two buoys. Are the values averaged? Are the forecasts then averaged between the two locations also?

Response: As the hurricane translates throughout the region, it may first be observed by buoy 'A', before being observed by buoy 'B'. Rather than averaging between the buoy locations and losing significant information, we chose to connect the time series together thereby increasing the total length of the dataset. Events like this were very rare and we thought it prudent to do this considering the severe lack of data in the first place. To ensure transparency, the dataset we used will be made available to interested parties.

Reviewer Comment 8: Section 3.2: This section would be clearer if an explanation were included of what constitutes an event in the SWH histograms presented. Is each event a particular time period? How does this relate to the earlier time-series?

Response: We have completely rewritten this section and have removed the concept of a SWH event. Instead, we focus on observations and forecasts of SWHs in bins of (excluding >8 m) 1 m.

Reviewer Comment 9: Figure 6: The legend and caption disagree on which colour is used for the observations and forecast.

Response: This was a typo and was corrected to read:
"Figure 6. Histograms of Hurricane Dorian observed (red) vs forecasted (blue) SWH (m) at the (a) 0-, (b) 3-, (c) 6-, and (d) 12-hour forecast horizons. Results for the 9-hour forecast are presented in Figure S1."

   Additionally, note that Figure 6 is now Figure 7.

Reviewer Comment 10: Figure 6: There are some events with SWH<2m. Why are these not seen in the time-series in Figure 3? Also, the time-series in Fig3a peaks at

~7m while Fig6a shows events up to the 9-10m bin. Line 214 also states that Hurricane Dorian has a maximum SHW of 8m, but Figure 6a-d shows events in the 9-10m range. The differences between these figures requires some explanation.

Response: In analyzing the code to produce those figures, we have noticed that both the way the data was binned, and labelling was at fault for the appearance of events with SWH<2 m. We have identified the bugs, eliminated them from the code and replotted. The new figures remain consistent with the text, but minus those SWH<2 m events. Additionally, the forecasts produced values within the 9 – 10 m range, but there are no corresponding observations to match, therefore demonstrating that the model over-estimated the wave height for each hurricane. We have also corrected the figures within the supplementary file.

Additionally, note that Figure 6 is now Figure 7.

Reviewer Comment 11: Lack of a metric to capture the timing of the peak SWH.

Response: We are unaware of a metric that can capture the timing of the peak SWH under hurricane conditions as this is generally not known a priori given the vast quantity of variables involved in this phenomenon. If you do know of a metric, we humbly ask that you inform us so that we can do the calculations that we think would be very valuable in increasing the impact of our work.

Reviewer Comment 12: Figures 3, 4, and 5 all show (to different extents) a phase shift in the timing of the peak SWH between the observations and forecasts. This appears to increase as the forecast leadtime increases. This requires some consideration and discussion as it appears to be a major drawback from this method of forecasting.

Response: This is an inherent problem with LSTM and has not yet been adequately resolved in existing literature. We have amended both the main text and discussion to take into consideration. We thank the reviewer for their attention to detail.

Reviewer Comment 13: I suggest including an additional metric to quantify the ability to capture the timing of the peak SWH. Perhaps a lagged correlation of the observations and forecasts would quantify how well the timing of the peak is captured

at different forecast leadtimes. Indeed, the lag between observations and forecast shown for some of the test data (e.g., Figure 5) appears to match the forecast leadtime.

Response: We have used autocorrelation to describe the lags that exist in the forecasts of Hurricanes Sandy and Igor, in addition to providing a reason for why these lags occur.

Reviewer Comment 14: Lack of a comparison method. The paper would be strengthened considerably from a comparison of this method to another. Even in broad terms, describing the approximate RMSE expected from wave model forecasts at similar leadtimes would be useful. This could help the reader understand the forecast leadtimes beyond which a wave model significantly outperforms this method.

Response: We have added another section (3.4) to characterize the difference between a wave model (SWAN) hindcast and LSTM focusing on the differences in model accuracy with regards to observations, required computational expense and expertise. We believe this would be best at this juncture.

Reviewer Comment 15: Alternatively, since this paper includes both SWH and surface wind speeds as input to the LSTM neural network, a comparison against forecasts using only SWH as input would allow readers to judge the benefit of the developments made by the authors.

Response: We thank the reviewer for this suggestion as it is indeed a very important component of the current work we are doing here. The issue of the relative effect on using either SWH or surface wind speeds on forecast efficacy is the subject of research that runs parallel to the current study and was actually borne out of it. We thought it prudent to devote a full paper to its discussion, rather than relegating it to a comparatively small discussion in the present manuscript. For now, readers may consult Wang et al. (2021) for an example of the usage of both wind speed and significant wave height for SWH predictions, but others focusing on just using SWH for predictions are available (e.g., Shamshirband et al., 2020; Zhou et al., 2021)

Wang, J., Wang, Y., Yang, J: Forecasting of SWH Based on Gated Recurrent unit Network in the Taiwan Strait and Its Adjacent Waters, Water, 13(1), 86, https://doi.org/10.3390/w13010086, 2021.

Zhou, S., Bethel, B.J., Sun, W., Zhao, Y., Xie, W., Dong, C: Improving SWH Forecasts Using a Joint Empirical Mode Decomposition-Long Short-Term Memory Network, J. Mar. Sci. Eng., 9, 744, 2021a.

Shamshirband, S., Mosavi, A., Rabczuk, T., Nabipour N, and Chau K: Prediction of significant wave height; comparison between nested grid numerical model, and machine learning models of artificial neural networks, extreme learning and support vector machines. Engineering Applications of Computational Fluid Mechanics, 14, 805-817, 2020. https://doi.org/10.1080/19942060.2020.1773932

**Specific comments**

Reviewer Comment 16: Figure 1: It would also be useful to highlight the test data tracks in particular to allow readers to judge the representativeness of the training data compared to the test data.

Response: In replotting Figure 1, we have added separate line colors for each of the test data tracks, while ensuring that the training test tracks are a uniform black. The corresponding caption was also modified to take into consideration the changes. We thank the reviewer for this suggestion.

Reviewer Comment 17: Figures 3, 4, & 5: It is difficult to judge the relative difference between observation and forecast between the panels. I suggest the authors consider keeping the top panel as it is (to show the range of SWHs), but for each forecast to show the observation-minus-forecast differences.

Response: We have taken the reviewer's suggestion into consideration and replotted Figures 3, 4, and 5 to show the differences between the observations and forecasts.

Reviewer Comment 18: Line 115: "data partitioned into a 70/30 split". I'm confused by this statement as this doesn't correspond to the number of hurricanes used in the training and validation sets.

Response: Here, we refer not to the number of hurricanes that were partitioned into the 70/30 split but following model training (using all of the data), the model was used in addition to 70% of a particular hurricane's data, and the remaining 30% is used for testing. We have provided a more complete explanation within the text at that location.

Reviewer Comment 19: Line 126: This is not the standard Pearson correlation coefficient. Although a value of 1 is returned if x=x dot, a value of -1 is not returned when x= -1(x dot). In fact, the return values are not restricted to the range -1 to 1. As such, it is difficult to interpret the meaning of the correlation coefficients presented.

Response: We have corrected that line and provided the standard Pearson correlation.

Reviewer Comment 20: Line 141: The authors state that the LSTM nowcast is unable to capture the extremely fine details seen in observations. I would welcome some discussion as to why this is the case. Are the observations noisy, or is there something fundamental to the LSTM which smooths the output compared to the inputs?

Response: In a previous study of ours (Zhou et al., 2021), we have tackled this problem specifically. This occurs because the model is unable to learn extreme fine details given their rarity and extremely short durations, and thus even after processing using empirical mode decomposition, the high-frequency components of the signal are completely missed. Your guess that the observations are noisy is accurate. After that sentence, we have added the reason for this problem and the reference to support it as follows:

This is because there are far too few examples of high-frequency components of the signal that the model could learn from and reproduce. Even following preprocessing using Empirical Mode Decomposition, high-frequency components of original SWH signals remain a challenge for LSTM (Zhou et al., 2021).

Zhou, S., Bethel, B.J., Sun, W., Zhao, Y., Xie, W., Dong, C: Improving SWH Forecasts Using a Joint Empirical Mode Decomposition-Long Short-Term Memory Network, J. Mar. Sci. Eng., 9, 744, 2021a.

Reviewer Comment 21: Line 161: "but this was minor" The difference referred to is approximately 2m, I do not agree that this is minor.

Response: After re-reading the text, we agree and have removed that portion of the sentence.

Reviewer Comment 22: Lines 197-199: The text here doesn't match what is shown in Figure 6a.

Response: We thank the reviewer for noticing this issue. We have made corrections as follows:

In Fig. 6a, it can be observed that for the 0-hour SWH nowcast, the model largely overestimated the frequency of wave heights within the 3 m bin, but skill improved remarkably with higher waves within the 4 m – 8 m bins. Unfortunately, the model also severely overestimated the number frequency of waves at the 1 m or 9 m range as there were no observations of waves at those heights.

Reviewer Comment 23: Line 198: "completely reproduce" While the correspondence is close, it is not exact. Please rephrase.

Response: We have amended that to read "largely reproduce" to better reflect the facts. We thank you for this suggestion.

Reviewer Comment 24: Line 201: "providing results for wave heights at the 8-9m range, though there are no observed occurrences". This doesn't agree with what is shown in Figure 3a which shows some events for both observations and forecast for all bins >3m.

Response: The text has been rewritten to conform with the new figures and as such, this line was removed entirely.

Reviewer Comment 25: Line 217: "At the 12hour horizon…completely missed". From the figures, I do not see any clear difference in this respect between panels a-d.

Response: We thank the reviewer for careful inspection of the figures. Following updating the figures, we have decided to remove that sentence entirely, and the preceding one to focus on the general feature of the model.

Reviewer Comment 26: Line 219-220: "model predicts wave heights that are approximately 1m higher than the total" Rather than "the total" I assume the authors mean "the model". Also, this seems to be an important point, but it is not clearly demonstrated by the current figures. I would suggest considering how best to support this point with an additional figure.

Response: We thank the reviewer for this suggestion and have provided a new figure for each hurricane to support the conclusion, placing them within the supplementary figures document. The discussion section was amended to discuss them.

Reviewer Comment 27: Line 221: "This may indicate…inclusion of less powerful hurricanes". Since the test data includes more powerful hurricanes than the training data, a conclusion that additional less powerful hurricanes should be added to the training set seems counter-intuitive. I would welcome some additional discussion on this.

Response: We thank the reviewer for the suggestion. Although the training dataset contains, in comparison with the test dataset, significantly less powerful hurricanes, maximum wind speed is not the only determinant in the wave heights a hurricane can generate. Indeed, the specific tracks, translation speed and environment (e.g., available fetch and duration), and confounding factors (e.g., sea surface currents) all have an impact on wave growth, maintenance, and decay. Thus, we advocate that less powerful hurricanes be added to the training set as a control (i.e., minimizing maximum wind speeds available to grow surface waves, regardless of environment or confounding factors) to decrease the probability of preferentially populating the training set with large surface waves. We have added this discussion, along with supporting references, at Line 221.

Reviewer Comment 28: Line 226: "previously not observed for either Hurricanes Dorian or Sandy". It is not clear to me from Figures 6, 7, and 8 that the behaviour is markedly different.

Response: We have considered the figures again and have revised the text by removing that line entirely.

Reviewer Comment 29: Line 227: "identical to the previous hurricanes, the frequency of maximum wave height predictions are overestimated". If the blue histogram shows the forecasts (unclear as caption and legend disagree) this is true for Fig8 and Fig7bcd, but not Fig7a or Fig6.

Response: These figures were updated and replaced, and their text modified to be in line with them. This issue no longer exists in the updated document.

Reviewer Comment 29: Figure 9: Although the text details the differences seen between the hurricanes in terms of R and RMSE (shown in Figure 9), I think some discussion of why the correlation for Sandy decreases faster than the other hurricanes is warranted.

Response: We have added a brief discussion as to why the correlation decreases faster for Sandy than for either Dorian or Igor.

Reviewer Comment 30: Lines 252-253: Stating these values in the text doesn't add anything, they are already shown clearly in the figures.

Response: Based on your suggestion, we have removed these sentences. The beginning of the paragraph now reads:

In Fig. 11, the MAPE for each of the hurricanes are given. There, it can be observed that Hurricane Dorian had MAPE values of 2.6% at the 0-hour nowcast and values of 7.99%, 10.83%, 13.13%, and 14.82% respectively at the 3-, 6-, 9-, and 12-hour forecast horizons.

Reviewer Comment 31: Lines 261-267: Again, this adds nothing to the paper. These values are all listed in a table and shown in a figure. It would be more useful to comment on the large difference in MAPE between Dorian and Sandy/Igor at long forecast leadtimes.

Response: We have removed those sentences and expanded upon the discussion on the large difference in MAPE between Dorian and Sandy/Igor at long forecast lead times. We continue the discussion as follows:

The difference in MAPE, in addition to the R and RMSE, may be due to the nature of Hurricane Dorian's time series of wave heights as the system approached NDBC buoy 41010 (Figure 1; Figure 3). Unlike Sandy or Igor where wave heights gradually grew to a peak and then declined, Hurricane Dorian's profile was far more gradual, allowing for LSTM to learn a comparatively much simpler pattern for forecasting. Indeed, unique to Hurricane Dorian, waves induced by the system were only observed after they would have affected and be affected by the Bahamas' continental shelf and its northern islands. Additionally, as is well understood, islands induce extensive modulation of the oceanic wave field. The presence of islands may cause modifications to wave spectra, reductions in wave heights, and triggering wave diffraction (Cao et al., 2018; Björkqvist et al., 2019; Passaro et al., 2021; Violante-Carvalho et al., 2021). As seen for Hurricane Joaquin (2015) by Sahoo et al. (2018), nonlinear wave setup and setdown processes occur when the system interacted with The Bahamas' varying coastal bathymetry, slope, and arching coastlines, and these, in conjunction with Hurricane Dorian's inherent properties (i.e., it's extremely slow translation speed of ~1.4 – 2 m/s), may have all played varying roles in the significantly lower variability in the pattern of wave growth at NDBC buoy 41010.

**Technical comments**

Reviewer Comment 32: Figure 1 uses a rainbow colour scale which is generally discouraged as it is not easily interpreted by colour-blind readers. I would encourage a change to a more accessible colour-scale.

Response: We thank the reviewer for this suggestion as it would greatly reduce the marginalization of those with colorblindness in the scientific enterprise. We have replotted Figure 1 using the 'viridis' colormap available in MATLAB. Unfortunately, we are unaware of anyone in our vicinity that is colorblind and thus we are unable to assess the efficacy of the change. If you know anyone that would be able to assist, we would be more than happy for additional feedback and any future requests to change the colorscale for their benefit.

Reviewer Comment 33: Figure 1: Triangles marking buoy locations are difficult to discern. Perhaps increase symbol size.

Response: We have increased the symbol size to ensure that readers can identify buoy locations. We thank you for the recommendation.

Reviewer Comment 34: Line 213: "Fig5a" I think the authors mean to refer to Fig 4a here.

Response: The reviewer is thanked for noticing this error. Fig 4a is indeed correct and this was changed within the article.

Reviewer Comment 35: Lines 235-238: This text appears to refer to all 3 hurricanes, but the paragraph continues on to Sandy and Igor, so this section must be specific to Dorian. Please clarify. I think it would be clearer to focus on the broad results of all 3 hurricanes rather than the detailed values of R, RMSE, etc for each (as these are in the figures anyway).

Response: The reviewer is thanked for their close attention to detail. You are correct in that the text should refer to all three hurricanes and as such, the text was amended as follows:

"Overall forecast quality can be assessed through the statistical metrics of R, RMSE, and MAPE, with results for each hurricane illustrated graphically in Fig. The full range of statistics is available in Table 3. In Fig. 10, it can be observed that regardless of hurricane, model forecast effectiveness (R) hovered near a perfect 1, but naturally deteriorated over time. By the 3-hr horizon, the three cases diverged from another in reflectance of each hurricane's characteristics. By the 12-hour horizon, the model was able to maintain accuracies above 0.8 in the majority of cases, which demonstrates that the model remained highly effective at predictions over a 12-hour time frame. Errors are also minimal: within a 6-hr forecast, RMSEs in all cases can be maintained under 1 m, but this is increases to just under 1.6 m after a further six hours. Thus, it is suggested that short-range 0 – 6 hour forecasts be prioritized over 12 hours when precision, rather than accuracy is required. Moreover, out of the hurricane cases, Hurricane Sandy's R performance decreased more rapidly than either Hurricanes Dorian or Igor. This may be related to the hurricane's track through the central

Caribbean Sea (Figure 1). There, both the Caribbean Low-Level Jet (CLLJ) and Caribbean Current flow in the atmosphere and ocean, respectively. It is thought that rather than Sandy's induced wave properties being affected by CLLJ which would have its normal zonal (with the main axis at 15°N) flows disrupted by the hurricane itself, the Caribbean Current would undoubtedly have changed hurricane-induced wave properties. Wave-current interactions have been widely demonstrated to change surface wave properties in a variety of scenarios including, but not limited to tidal flows (Hopkins et al., 2015), large-scale current structures such as the Loop Current and eddies (Romero et al., 2017), but as relevant for this discussion, also hurricane-induced wave interactions with large-scale currents (Sun et al., 2018; Hegermiller, et al., 2019). Unfortunately, as NDBC buoy 42058 that measured the passing of Sandy does not possess surface current information, this hypothesis cannot be tested using the available dataset or possible wave-current effects on hurricane wave field prediction quantified. The rapid decrease in R observed for Sandy could possibly be related to surface current-induced changes in the wave field not accounted for by the dual usage of wind speed and wave height as LSTM predictors for the wave height predictand."

Reviewer Comment 36: Line 245: "decreased to 0.82" I think this is the value for Dorian, not Igor (which is being discussed at this point in the text).

Response: We have examined the table again, and this value is correct. Dorian's value is a very similar at 0.84.

Reviewer Comment 37: Figure 9: It is difficult to distinguish the symbols used for Sandy and Dorian. Perhaps an open square for one would be better, or simply increasing the symbol size throughout.

Response: To improve the ability for readers to distinguish between Sandy and Dorian, we have changed Figure 9 by increasing the symbol sizes. We have also changed Figure 10 for the same problem. Note that these figures are now Figures 10 and 11 in the updated manuscript.

Reviewer Comment 38: Figure 9 caption refers to only one hurricane while 3 are shown in the figure.

Response: Thank you for noticing this issue, you are correct. The caption was amended to read:

"Figure 9. LSTM model forecast performance in terms of R (blue) and RMSE (red) as compared with the observations for Hurricanes Dorian, Sandy, and Igor."

Also note that this figure is now Figure 10 in the updated manuscript.

Reviewer 2 Comments

This paper proposes the use of a Long Short-Term Memory architecture to predict hurricane-forced significant wave heights in the Caribbean Sea. The application is interesting, and the use of this model is motivated by the necessity of having fast predictions.

Reviewer Comment 1: The data is provided by 10 buoys and the number of scenarios (as shown in table 2) is not big which may represent a challenge for the generality of the model. The choice of merging multiple observations in one time series (lines 75-76) is particularly important for preventing overfitting but the separation between training and testing is not motivated. Probably a better description of Figure 1 may be used help with this.

Response: We thank the reviewer for their close inspection of the manuscript. In response to another reviewer, we have updated Figure 1 to better demarcate the training and test cases from each other. Here, we have added the following text so that readers can have a better understanding of the motivation for the cases chosen for our test cases:

"Training sets were chosen to represent hurricanes that passed through the study area (Figure 1) over the 2010 – 2019, ten-year period to ensure that the model is sufficiently general to be used, regardless of prevailing environmental conditions/phenomena which may influence hurricanes of a particular year (e.g., El Niño/La Niña). Test sets, by contrast, were chosen to be extremely powerful systems as it is highly expected that TCs, due to anthropogenic climate change, would gradually grow in intensity and frequency over time. Hurricane Dorian, in particular, was chosen given the widescale destruction it caused in The Bahamas and will be used in a comparison with numerical model output in Section 3.4."

Reviewer Comment 2: The model proposed is a standard LSTM. This recurrent neural network is well known to be good in training time series. The choice of the

hyperparameters is not described or motivated. The only information is about the number of epochs and the batch size but there is not mentioned any reason for these choices and there are not comparisons with others.

Response: We have updated the description of the LSTM used and corrected an error that was present in the original manuscript. We have chosen those parameters following experiments but not save the results for display within the article. We have provided references of our recent using similar settings to justify the applicability of those parameters. That section was updated as follows:

"LSTM is set up with four layers that correspond to a time step of four. The recursive linear unit (ReLu) was used as the activation function to maximize the model's ability to capture nonlinearities. The Adaptive Moment Estimation (Adam) optimizer is used to compute adaptive learning rates. The number of epochs was set to 100 and the batch size set to 3. Throughout each experiment, the operating parameters were held constant. These settings were chosen after experiments (not shown) as they produced the best results while avoiding overfitting. Similar settings can be found in Bethel et al. (2021a) and Zhou et al. (2021a, 2021b). The data was partitioned along a 70/30 split into training and validation datasets. For clarification, here, and only here, the word 'dataset' should be interpreted as a given test hurricane (the test set hurricanes of Table 2). A general model is trained using the training set hurricanes of Table 2, but the model is specified to a given test set hurricane using 70% of its time series, and the remaining 30% is used to validate the forecast. "

Reviewer Comment 3: The results in Figures 3,4,5 show some discrepancies between the forecasting and the observation. In case the authors are willing to, this may be solved by implementing an adversarially-trained LSTM.

Response: We thank the reviewer for close attention. Results in Figures 3, 4, and 5 were incorrectly plotted and were replotted which, excluding the LSTM-inherent phase shifting phenomenon issue, resolved this issue without needed to use an adversarially-trained LSTM. The phase shifting, most notable in Hurricane Igor, is a problem with LSTM itself and remains an open problem in the mathematics underlying this network. That being said, we would be happy to try the usage of an

adversarially-trained LSTM in a future problem addressing the phase shifting in hurricane predictions and thank the reviewer for the suggestion.

Reviewer Comment 4: The authors provide results of the accuracy. However, it would be interesting to check the efficiency, at least in terms of execution times for both training-testing and for the forecasting.

Response: Following a few minutes of model training, forecasting occurred in a very small fraction of a second. We have added this feature of the model in Section 3.4 when we discuss the model in comparison with a numerical model (SWAN). We thank the reviewer for the suggestion.

Reviewer Comment 5: As the authors state in the introduction (from lines 29), there are other methods for nowcasting which are actually used for these purposes. The paper may provide comparisons in terms of accuracy or/and efficiency with these methods at least in terms of order of magnitude.

Response: We thank the reviewer for the suggestion. Another reviewer has also made the same suggestion and thus, we introduce Section 3.4 to compare LSTM output with a SWAN numerical model hindcast from the perspective of accuracy, computational requirements and required level of technical expertise.

---

## Referee Report (RR1)

I would like to thank the authors for their significant efforts in revising this manuscript. The issues I raised regarding the confusion between some text and figures, the construction of the observed time-series when using multiple buoys, and issues with figures have all been addressed. I think the additional model comparison section and discussion of the lagged correlations inherent in the LSTM method significantly enhance the paper.

**Minor comments**

Original comment #2: Additionally, from Table 2 the test date have generally deeper lows and faster wind speeds than the training set. Some discussion on the use of this method on hurricanes which fall at the extremes, or outside the range of conditions in the training set would be beneficial.

Author Response: We agree and have added the following discussion at Line 492:

"Thirdly, the selection of training and test sets would have an extremely strong impact on forecasting results. Specifically, test sets of this study (i.e., Hurricanes Dorian, Sandy, and Igor) were are all far more powerful than hurricanes within the training set, and these were chosen as it is expected that due to climate change, hurricanes are due to not only become more frequent, but also, more intense. The present method demonstrates that the model overestimates the highest SWHs of even those systems and should continue be effective if hurricanes become even more extreme (and thus, the degree by which the current model overestimates maximum SWHs should decrease). However, if future systems are weaker than the test set (as it is now), the problem of overestimation would be exacerbated. Thus, a second model that is trained with hurricanes even weaker than the training set would be prudent and run in parallel to ensure both scenarios are considered in future disaster aversion strategies."

**Reviewer response:**

I think the final sentence of the new text conflates two issues: firstly, the difference in character of the training set and trial set hurricanes, and secondly the tendency of the method to overestimate maximum SWHs. While it is instructive that in this paper the model is applied to more powerful hurricanes than are included in the training set, subject to correcting the method's bias to larger SWHs, I think it would be prudent to use all of the historical hurricane data available in a training set before the model is used operationally.

**Original Comment #11: Lack of a metric to capture the timing of the peak SWH.**

Author Response: We are unaware of a metric that can capture the timing of the peak SWH under hurricane conditions as this is generally not known a priori given the vast quantity of variables involved in this phenomenon. If you do know of a metric, we humbly ask that you inform us so that we can do the calculations that we think would be very valuable in increasing the impact of our work.

Reviewer response: My original comment was intended to suggest comparing the SWH maxima seen in your observed and forecasted time-series presented in Figures 3, 4 & 5. Your new figure 6 and response to comments #12 and #13 addressed this.

Reviewer Comment 32: Figure 1 uses a rainbow colour scale which is generally discouraged as it is not easily interpreted by colour-blind readers. I would encourage a change to a more accessible colour-scale.

Response: We thank the reviewer for this suggestion as it would greatly reduce the marginalization of those with colorblindness in the scientific enterprise. We have replotted Figure 1 using the 'viridis' colormap available in MATLAB. Unfortunately, we are unaware of anyone in our vicinity that is colorblind and thus we are unable to assess the efficacy of the change. If you know anyone that would be able to assist, we would be more than happy for additional feedback and any future requests to change the colorscale for their benefit.

Reviewer response: Thank you for making this change. I believe that 'viridis' is a much more accessible choice. There are some web-sites which allow you to simulate the effect of various forms of colour blindness on your own figures, e.g., https://www.color-blindness.com/coblis-color-blindness-simulator/